

# Sailing past the End of the World and discovering the Island

Tarek Anous[1*], Marco Meineri[2†], Pietro Pelliconi[2‡] and Julian Sonner[2§]

**1** Institute for Theoretical Physics and Δ-Institute for Theoretical Physics,
University of Amsterdam, Science Park 904, 1098 XH Amsterdam, The Netherlands
**2** Department of Theoretical Physics, University of Geneva,
24 quai Ernest-Ansermet, 1211 Genève 4, Suisse

\* t.m.anous@uva.nl , † marco.meineri@gmail.com ,
‡ pietro.pelliconi@unige.ch , § julian.sonner@unige.ch

## Abstract

Large black holes in anti-de Sitter space have positive specific heat and do not evaporate. In order to mimic the behavior of evaporating black holes, one may couple the system to an external bath. In this paper we explore a rich family of such models, namely ones obtained by coupling two holographic CFTs along a shared interface (ICFTs). We focus on the limit where the bulk solution is characterized by a thin brane separating the two individual duals. These systems may be interpreted in a double holographic way, where one integrates out the bath and ends up with a lower-dimensional gravitational braneworld dual to the interface degrees of freedom. Our setup has the advantage that all observables can be defined and calculated by only relying on standard rules of AdS/CFT. We exploit this to establish a number of general results, relying on a detailed analysis of the geodesics in the bulk. Firstly, we prove that the entropy of Hawking radiation in the braneworld is obtained by extremizing the generalized entropy, and moreover that at late times a so-called 'island saddle' gives the dominant contribution. We also derive Takayanagi's prescription for calculating entanglement entropies in BCFTs as a limit of our ICFT results.



# 1   Introduction and summary

The AdS/CFT correspondence offers a well-controlled theoretical laboratory to study quantum gravity. Some of the most interesting issues concern the evaporation dynamics of black holes. However, in trying to address this within the framework of holography we need to confront the technical obstacle that (large) black holes in asymptotically Anti de Sitter spaces coexist in equilibrium with their Hawking radiation and consequently do not evaporate [1]. In order to have access to the evaporation dynamics one therefore needs to couple the black hole system to a bath and understand their combined dynamics (see e.g. [2,3] for an early exposition). More recently, such setups have been reconsidered, leading to a new semiclassical understanding of the evaporation dynamics of black holes, which are notably brought in accord with unitary expectations [4,5] due to the appearance of a new semiclassical saddle at late times that acts to unitarize the radiation, [6,7].

However, just like the original Euclidean path-integral approach of black-hole entropy due to Gibbons and Hawking [8] does not capture the underlying microstates, the revised semi-classical approach does not provide a microscopic understanding of the evaporation dynamics. In order to make progress in this direction one needs to understand the interplay of systems and bath degrees of freedom at a more fine-grained level. One class of candidate systems in which such a study can be performed are boundary conformal field theories, [9–15]. Here we would like to interpret the boundary degrees of freedom as 'the system', while the bulk conformal field theory (CFT)[1] takes on the role of 'the bath'. In particular, the authors of [14,15]

---

[1]We should at this point make a comment on terminology: in this work there are (at least) two types of boundaries we need to refer to. There is the conformal boundary of the asymptotically AdS space, and there is the boundary/interface of the holographically dual CFT. Which one is meant should be clear from the context. Furthermore, there is the *interior* of the AdS space, to be distinguished from the *bulk* of the CFTs, the latter denoting everything that is not either the boundary of the interface between the CFTs we consider.

extended this setup by considering degrees of freedom supported on a defect which splits the bulk CFT into two distinct parts, rather than bounding it at the edge, and they highlighted some conceptual advantages of this approach. In this work we add a much larger class of well-controlled systems by instead engineering boundary ('system') degrees of freedom located at the interface between two different CFTs, and use the bulk of both CFTs together as the 'bath degrees of freedom'. We thus study a class of interface CFTs (ICFTs), focussing in particular on those with holographic duals.[2] We first establish a number of general results on correlation functions in ICFTs before presenting a number of applications, including to studying Page curves of evaporating black holes, where we establish that the 'island' prescription in the ICFT setup arises naturally from considering standard AdS/CFT rules for the computation of entanglement entropies. Furthermore, we also describe how, by varying the difference of central charges across the interface, many results on boundary CFT (BCFT) arise as natural limits of our more general setup, giving derivations of previous results while relying only on standard tools and techniques for computing correlations functions and entanglement entropies in AdS/CFT.

We now outline some further aspects of our ICFT systems, which can be interpreted in a number of different ways. As illustrated in Figure 1, the double holographic setup consists of three descriptions of the same system. The ICFT and its holographic dual are UV complete and stand on equal footing. The gravitational description contains a brane—an end-of-the-world (EOW) brane in the special case of a BCFT [17, 18]—which is forced to end on the interface. In the semi-classical approximation, a third description of the system emerges. The brane itself can be interpreted as a weakly gravitating universe, coupled to the bulk CFT through the interface. This effective theory provides the 'system+bath' setup mentioned above. Recent progress in understanding black-hole evaporation came from computing the same observable in either of the three descriptions [10, 11, 13, 14, 19, 20]. The subject of lower-dimensional AdS-branes within AdS and their induced braneworld gravity has a long history, going back to [21, 22].

One set of results of the present paper demonstrates the advantages of this perspective in concrete examples. We now give a preview of the intuition one can gain from taking a careful look at what quantities can be matched on the three sides. Consider a two dimensional BCFT, and compute the entanglement entropy in the vacuum of an interval which includes the defect and part of the bath. The Takayanagi prescription in AdS/BCFT instructs us to compute the length of the minimal geodesic which stretches between the entangling surface and a point on the brane. The result matches the CFT computation, and can be derived as a consequence of the ordinary Ryu-Takayanagi formula, as we elucidate in section 5. But what is the rule in the two dimensional 'system+bath' description? One might think that the full gravitational region must be traced over, as was the boundary point in the CFT computation. This intuition would be wrong, as it becomes clear in the large tension limit. The EOW brane is pushed outwards and reconstructs the missing piece of the conformal boundary of $AdS_3$. Therefore, the dynamics of matter on the brane becomes conformal in this limit. The Takayanagi prescription for the entanglement entropy in a BFCT now reduces to the Ryu-Takayanagi rule for the 'system+bath'. We are led to conclude that the correct result is reproduced by the entanglement entropy of an interval which includes only part of the weakly gravitating region. How is this second end-point determined from the point of view of the two-dimensional observer? Since Takayanagi's formula included a minimization step, we show that one is automatically lead to the celebrated island rule.[3] We see that even when gravity is arbitrarily weakly coupled, it still

---

[2]Such an ICFT setup with applications to entanglement islands, composed of free fermions, was considered in [16].

[3]One might ask why the minimization does not lead to an arbitrarily short interval. The reason is that the gravitating region hosts a conformal field theory on an $AdS_2$ background, where the AdS boundary coincides with the defect: the large curvature close to the boundary is responsible for the non-trivial minimum.

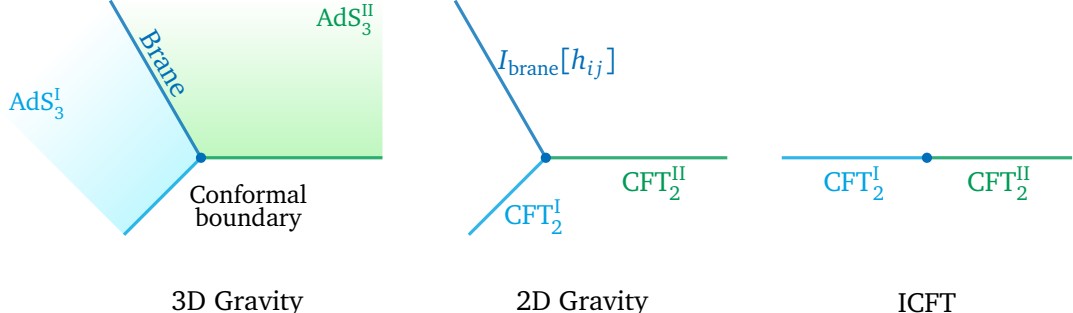

Figure 1: Three different perspectives on the same double holographic system. *Left:* the 3D bulk perspective, in which the two AdS spaces are the holographic duals of the CFTs and the thin brane is the holographic dual of the quantum dot. *Center:* the intermediate perspective, in which we see the brane as a gravitational system coupled to two non–gravitating baths, the CFTs. The gravitational theory on the brane, with action $I_{\text{brane}}[h_{ij}]$, is the dual of the quantum dot, and it is obtained by integrating out the AdS$_3$ bulk degrees of freedom. *Right:* the boundary perspective, in which we consider only the two CFTs and the quantum mechanical system which couples them.

has an order one effect on the structure of entanglement: a striking fact, which is remarkably within the reach of standard AdS/CFT.

The remainder of this paper is structured as follows. In section 2 we introduce the setup of our system, and describe in detail how holographic interface CFTs constitute a well-controlled 'doubly holographic' scenario. In section 3 we embark on a detailed study of the geometry of geodesics in the bulk dual of holographic ICFT setups, which in turn help us to determine a range of correlations functions and entanglement entropies of interest. What follows in section 4 is one of the main sets of results of this paper: by computing the entanglement entropy of early and late radiation from various different perspectives, we provide a derivation of the quantum-extremal surface prescription in our case. Section 5 is dedicated to a detailed derivation of the rules of AdS/BCFT by approaching these as a limit of AdS/ICFT (where the standard AdS/CFT dictionary applies). We conclude in section 6 and discuss some open issues.

*Note: as this paper was nearing completion, the preprint [23] appeared on the arXiv, which overlaps with some of our results, particularly regarding the correspondence between holographic BCFTs and the island formula in braneworld holography. Our paper focuses on holographic ICFTs, which include BCFTs as a limiting case, as we explain in section 5. A preliminary comparison suggests that the BCFT limit of our results are in agreement with those of [23].*

## 2 Double holography for Interface CFTs

### 2.1 Review of ICFTs

Physical systems at criticality are the realm of conformal field theory. We can probe these systems by measuring their response to local excitations. Alternatively, we can turn on couplings along extended submanifolds, and measure their effect on observables. A natural possibility is to couple two different critical systems along a mutual boundary. If the latter is conformal invariant, the combined system is known as an *interface conformal field theory* (ICFT), and has been the subject of many papers over the years, both in condensed matter [24–35], and

in holography [22, 36–44]. Semi-transparent interfaces can be engineered in various ways. An obvious possibility is to first pick conformal boundary conditions for two $d$-dimensional CFTs—always denoted as CFT$_\text{I}$ and CFT$_\text{II}$ in the following. A conformal boundary condition, or BCFT, is defined by the property that observables are constrained by the subgroup of the conformal symmetry which preserves the (flat) boundary. If there are two boundary operators $\widehat{\mathcal{O}}_\text{I}$ and $\widehat{\mathcal{O}}_\text{II}$ respectively, such that their scaling dimensions obey $\widehat{\Delta}_\text{I} + \widehat{\Delta}_\text{II} < d - 1$, then one can turn on the coupling

$$\lambda \int_\text{boundary} \widehat{\mathcal{O}}_\text{I} \widehat{\mathcal{O}}_\text{II} \, . \tag{2.1}$$

Generically, we expect observables at large distances to be again constrained by conformal symmetry. However, rather than a tensor product of boundary conditions, the interface might be permeable and show non-vanishing correlations between the two sides.

The simple example (2.1) could be complicated in various ways: one can couple CFT$_\text{I}$ and CFT$_\text{II}$ indirectly, via lower dimensional matter localized on the interface, or consider multi-parameter flows. If the two CFTs coincide, one can construct defects by integrating bulk local operators on a codimension one surface [27]. It is also possible to tune a marginal coupling to different values on two half-spaces, or flow via a relevant deformation on half of the space, thus constructing Janus [36, 45] and renormalization group (RG) [33] interfaces.

In this paper, we shall mostly consider the case of two dimensional CFTs, although the qualitative picture easily generalizes, and we expect many specific results to be the same. The CFTs on either side of the interface are characterized by central charges $c_\text{I}$ and $c_\text{II}$. We will be interested in the case where both CFTs are holographic, so in particular $c_\text{I/II} \gg 1$ and both theories have a sparse spectrum. The holographic dual of this setup will consist of two asymptotically AdS$_3$ spacetimes with the AdS radii determined by the central charges of the two CFTs:

$$c_\text{I,II} = \frac{3 L_\text{I,II}}{2 G_{(3)}} \, . \tag{2.2}$$

The full bulk spacetime interpolates between these two AdS$_3$'s as we move along a spatial direction, and we will work in the approximation that the region connecting these geometries is simply a thin brane, as in [17, 39, 41, 43, 44, 46]. This setup readily generalizes the BCFT setup of [17, 47]. The precise geometry will be reviewed in subsection 2.3.

When the ratio $c_\text{I}/c_\text{II}$ is generic, the brane allows the trasmission of energy across the interface [31, 35, 42]. However, in the limit $c_\text{I} \ll c_\text{II}$, it becomes impossible for generic waves built from the CFT$_\text{II}$ degrees of freedom to scatter into the CFT$_\text{I}$. Thus, by this simple reasoning we obtain the physics of BCFT as a limit of ICFT. This will guide our story.

## 2.2 Changing faces of two-faced geometries

As announced, our setup consists of two AdS regions glued together at the location of a brane, as shown in the first panel of figure 1. The three interpretations of the system were referred to in the introduction and are explained in the caption of the same figure. This doubly holographic interface allows us to explore a variety of physical questions [39, 42–44], some of which we shall consider in detail in the rest of the paper.

One of the main reasons of interest is the same highlighted in the recent literature [9, 10, 14, 15, 48]. The quantum dot which separates the two CFTs has finite, albeit possibly large, entropy. If the system is put in a generic time dependent state, energy and entropy will be exchanged with the baths, until equilibrium is reached, see figure 2. The time evolution of the entanglement between the dot and the baths must be compatible with the finiteness of the Hilbert space of the defect, a fact which is the cornerstone of the Page curve [4, 5].

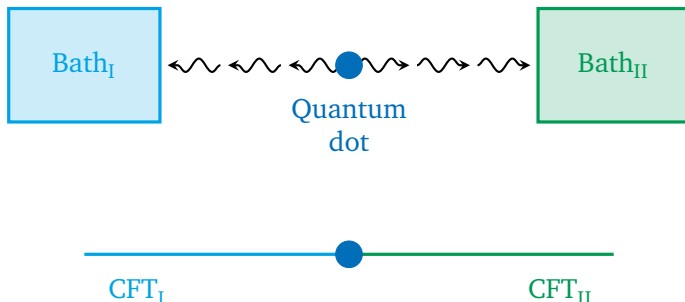

Figure 2: A possible physical interpretation of the system: a quantum dot coupled to two baths. The quantum dot has finite entropy, the boundary entropy of the defect, and if we put the system in the thermofield double state the dot can be seen as an eternal black hole coupled to two reservoirs.

While this does not imply a paradox yet, one can choose a state—the thermofield double— and a Hamiltonian—the difference of the global time translations on the two sides—such that stationary observers see a horizon on the brane, in the intermediate description of figure 1. What is more tantalizing, the system is under analytic control in the large $N$ limit, and the Page curve can be computed via the Ryu-Takayanagi formula. In section 4 we will expand on this topic.

Of course, ICFTs are interesting in their own right, and the possibility of exact computations at strong coupling extends beyond the thermofield double, where the black-hole arises, and also beyond entanglement entropy altogether. General two-point functions of single-trace heavy operators can be computed via the geodesic approximation. They are not fixed by symmetry, and they do not decouple from the interface. In the conformal bootstrap language, they exchange an infinite number of conformal blocks, despite being fixed by a single geodesic stretching between two boundary points. In section 3, we describe in detail how to compute these geodesics, and illustrate a variety of scenarios depending on the tension of the parameters of the ICFT and the position of the local operators. The resulting structure is quite rich. In particular, when the operators are thought of as twist fields, the results express the entanglement of subregions of the two CFTs. Via entanglement wedge reconstruction, they could shed light on the way a local bulk is encoded in the boundary, when the state is strongly perturbed away from the vacuum.

## 2.3 Bottom-up model

Our focus in this paper will be on interface CFTs dual to semiclassical gravity in AdS$_3$, particularly a geometry described by a thin brane separating two locally AdS$_3$ geometries with respective curvatures scales $L_{\text{I/II}}$ coupled through a permeable membrane. The Euclidean gravity action describing our system on $\mathcal{M}_{\text{I}}$ and $\mathcal{M}_{\text{II}}$ is:

$$
S_{\text{EH}} = -\frac{1}{16\pi G_{(3)}} \left[ \int_{\mathcal{M}_{\text{I}}} \mathrm{d}^3 x \sqrt{g_{\text{I}}} \left( R_{\text{I}} + \frac{2}{L_{\text{I}}^2} \right) + \int_{\mathcal{M}_{\text{II}}} \mathrm{d}^3 x \sqrt{g_{\text{II}}} \left( R_{\text{II}} + \frac{2}{L_{\text{II}}^2} \right) \right.
$$
$$
\left. + 2 \int_{\mathcal{S}} \mathrm{d}^2 y \sqrt{h} \, (K_{\text{I}} - K_{\text{II}}) - 2T \int_{\mathcal{S}} \mathrm{d}^2 y \sqrt{h} \right] + \text{corner and counterterms}, \quad (2.3)
$$

where $T$ represents the tension of the brane, $h_{ab}$ is the induced metric along it, and the extrinsic curvatures $K_{\text{I,II}}$ are computed with outward normal pointing from I→II in both cases. For details on the corner term and counterterms see for instance appendix A of [44].

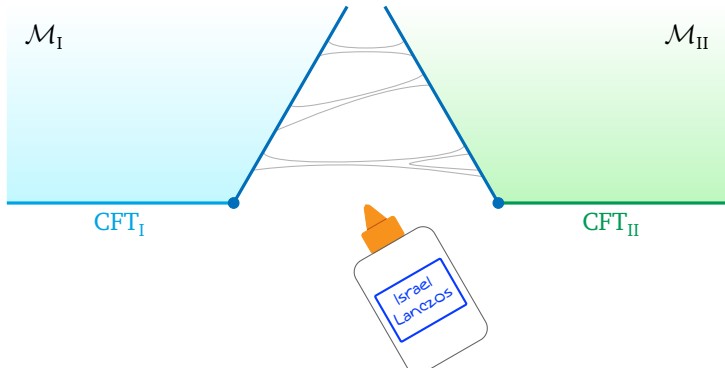

Figure 3: The bulk geometry for a linear defect in $AdS_3/CFT_2$. The two dark blue lines represent the brane, and the two spaces are glued with the Israel–Lanczos matching conditions.

The location of the brane between the two spacetimes $\mathcal{S} \equiv \partial \mathcal{M}_I \cap \partial \mathcal{M}_{II}$ is determined by the two Israel–Lanczos matching conditions [49, 50]. These matching conditions follow from the equations of motion of the action (2.3), and the saddle point approximation instructs us that they represent a reliable approximation of the position of the brane in the limit

$$\frac{T L_{I,II}^2}{G_{(3)}} \gg 1 \,. \tag{2.4}$$

This is satisfied in the usual regime $c_{I,II} \sim L_{I,II}/G_{(3)} \gg 1$, and assuming $T L_{I,II}$ an order one number. The solution of the equations of motion will suggest that the latter is generally respected, since it will enforce the constraint (2.30).

If $x_I(y)$ is the embedding of the brane in $\mathcal{M}_I$ and $x_{II}(y)$ is its embedding in $\mathcal{M}_{II}$, the first matching condition imposes equality of $h_{ab}$, as viewed from either spacetime:

$$h_{ab} \equiv \frac{\partial x_I^\mu}{\partial y^a} \frac{\partial x_I^\nu}{\partial y^b} g_{\mu\nu}^I = \frac{\partial x_{II}^\mu}{\partial y^a} \frac{\partial x_{II}^\nu}{\partial y^b} g_{\mu\nu}^{II} \,. \tag{2.5}$$

The second matching condition involves the two extrinsic curvatures $K_{i,ab}$, and requires that their discontinuity across the brane be proportional to the tension $T$. In terms of

$$\Delta K_{ab} \equiv K_{I,ab} - K_{II,ab} \,, \tag{2.6}$$

the second matching condition is

$$\Delta K_{ab} - h_{ab}\Delta K = -T h_{ab} \,. \tag{2.7}$$

Using the trace of (2.7), we can simplify the above expression in three dimensions[4] to:

$$\Delta K_{ab} = T h_{ab} \,. \tag{2.8}$$

## 2.4 Bulk geometry and coordinates

On either side of the brane, the spacetime will be locally $AdS_3$. Everything about the geometry of (Euclidean-)$AdS_3$ can be surmised by thinking of it as a hyperboloid

$$-\left(X^0\right)^2 + \left(X^3\right)^2 + \sum_{i=1}^2 \left(X^i\right)^2 = -L^2, \tag{2.9}$$

---

[4] The brane is a codimension one surface, so in three dimensions $h_{ab}h^{ab} = 2$.

embedded in four-dimensional flat Minkowski spacetime:

$$G_{\mu\nu}\mathrm{d}X^{\mu}\mathrm{d}X^{\nu} = -\left(\mathrm{d}X^{0}\right)^{2} + \left(\mathrm{d}X^{3}\right)^{2} + \sum_{i=1}^{2}\left(\mathrm{d}X^{i}\right)^{2} . \tag{2.10}$$

The metric of $\mathrm{AdS}_3$ can be obtained by finding a set of coordinates that 'solve' (2.9). One such set are the *global coordinates*,

$$X^{0} = L\sqrt{1 + \frac{r_g^2}{L^2}}\cosh\left(\frac{\tau_g}{L}\right) , \qquad\qquad X^{1} = r_g\cos\theta ,$$

$$X^{3} = L\sqrt{1 + \frac{r_g^2}{L^2}}\sinh\left(\frac{\tau_g}{L}\right) , \qquad\qquad X^{2} = r_g\sin\theta . \tag{2.11}$$

In this coordinates we have the local form of the metric

$$\mathrm{d}s^{2} = \left(1 + \frac{r_g^2}{L^2}\right)\mathrm{d}\tau_g^2 + \frac{\mathrm{d}r_g^2}{\left(1 + \frac{r_g^2}{L^2}\right)} + r_g^2\mathrm{d}\theta^2 . \tag{2.12}$$

The spatial boundary of $\mathrm{AdS}_3$ is the location where $\left(X^1\right)^2 + \left(X^2\right)^2 \to \infty$, and in these coordinates:

$$\left(X^1\right)^2 + \left(X^2\right)^2 = r_g^2 . \tag{2.13}$$

Thus we conclude that the spatial boundary coincides with $r_g \to \infty$, as expected.

An alternative set of coordinates, one that is particularly useful set for solving the junction conditions described above, is the $\mathrm{AdS}_2$ slicing of $\mathrm{AdS}_3$:

$$X^{0} = \frac{L^2 + \tau^2 + y^2}{2y}\cosh\left(\frac{\rho}{L}\right) , \qquad X^{1} = L\sinh\left(\frac{\rho}{L}\right) ,$$

$$X^{3} = \frac{L\tau}{y}\cosh\left(\frac{\rho}{L}\right) , \qquad\qquad X^{2} = \frac{L^2 - \tau^2 - y^2}{2y}\cosh\left(\frac{\rho}{L}\right) , \tag{2.14}$$

where $-\infty < \rho < \infty$ and $y \geq 0$ . This gives rise to the local form of the metric

$$\mathrm{d}s^{2} = \mathrm{d}\rho^2 + L^2\cosh^2\left(\frac{\rho}{L}\right)\left(\frac{\mathrm{d}\tau^2 + \mathrm{d}y^2}{y^2}\right) , \tag{2.15}$$

which is related to the standard Poincaré slicing of $\mathrm{AdS}_3$ through the following coordinate transformation

$$z = \frac{y}{\cosh\left(\frac{\rho}{L}\right)} , \qquad x = y\tanh\left(\frac{\rho}{L}\right) , \tag{2.16}$$

with metric

$$\mathrm{d}s^{2} = L^2\frac{\mathrm{d}\tau^2 + \mathrm{d}x^2 + \mathrm{d}z^2}{z^2} . \tag{2.17}$$

Either by looking at the embedding space coordinates $\left(X^1\right)^2 + \left(X^2\right)^2$ or the Poincaré coordinates, we note that the boundary of $\mathrm{AdS}_3$ at $z = 0$ can be reached either by taking $|\rho| \to \infty$ or $y \to 0$. Although not immediatly obvious, $\rho$ actually parametrizes an angular coordinate. Defining:

$$\tanh\left(\frac{\rho}{L}\right) \equiv \sin\chi , \tag{2.18}$$

then the Poincaré coordinates are simply:

$$z = y\cos\chi , \qquad x = y\sin\chi, \tag{2.19}$$

thus $y$ is a radial coordinate in the $xz$-plane. The asymptotic boundary can then be thought of as the locus $\chi = \pm\pi/2$ and $y$ then becomes the coordinate along the boundary.

Finally, AdS$_3$ is special because there exists a parametrization that induces a black hole horizon on the hyperboloid. These coordinates are

$$X^0 = L\,\frac{r_b}{r_H}\,\cosh\left(\frac{r_H}{L}\theta\right)\,, \qquad\qquad X^1 = L\,\sqrt{\frac{r_b^2}{r_H^2}-1}\,\cos\left(\frac{r_H\tau_b}{L^2}\right)\,,$$

$$X^3 = L\,\sqrt{\frac{r_b^2}{r_H^2}-1}\,\sin\left(\frac{r_H\tau_b}{L^2}\right)\,, \qquad\qquad X^2 = L\,\frac{r_b}{r_H}\,\sinh\left(\frac{r_H}{L}\theta\right)\,, \qquad (2.20)$$

leading to the following metric for the BTZ black hole:

$$\mathrm{d}s^2 = \left(\frac{r_b^2-r_H^2}{L^2}\right)\mathrm{d}\tau_b^2 + \left(\frac{r_b^2-r_H^2}{L^2}\right)^{-1}\mathrm{d}r_b^2 + r_b^2\mathrm{d}\theta^2\,. \qquad (2.21)$$

It is evident from (2.20) that regularity of the Euclidean-BTZ metric will require $\tau_b$ to be periodic with periodicity $\tau_b \sim \tau_b + \frac{2\pi L^2}{r_H}$. Moreover, choosing $r_H = iL$ gives us back a double analytic continuation of (2.11), with the roles of $X^0$ and $X^1$ swapped.

It is possible to go from Poincaré slicing (2.15) to global coordinates (2.12) via the coordinate transformation:

$$ (2.22) $$

and similarly, we can obtain the black hole metric (2.21) from Poincaré slicing using:

$$\rho = L\sinh^{-1}\left[\cos\left(\frac{r_H\tau_b}{L^2}\right)\sqrt{\frac{r_b^2}{r_H^2}-1}\,\right]\,,$$

$$\tau = e^{\frac{r_H}{L}\theta}\sin\left(\frac{r_H\tau_b}{L^2}\right)\sqrt{1-\frac{r_H^2}{r_b^2}}\,, \quad y = e^{\frac{r_H}{L}\theta}\sqrt{1-\left(1-\frac{r_H^2}{r_b^2}\right)\sin^2\left(\frac{r_H\tau_b}{L^2}\right)}\,. \qquad (2.23)$$

Note that this crucially means that a solution to the junction conditions in one set of coordinates can be brought into a solution in another set of coordinates via a coordinate transformation. Having set out some geometric basics and defined a number of convenient systems of coordinates we will now move on to actually solving the junction conditions.

## 2.5 Solving the junction conditions

The simplest way to solve the junction conditions is to work with the coordinates (2.15), that is we will take the coordinates on each side of the brane to be

$$\mathrm{d}s^2_{\mathcal{M}_i} = \mathrm{d}\rho_i^2 + L_i^2\cosh^2\left(\frac{\rho_i}{L_i}\right)\left(\frac{\mathrm{d}y_i^2+\mathrm{d}\tau_i^2}{y_i^2}\right) \qquad (2.24)$$

for $i = \{\mathrm{I}, \mathrm{II}\}$. From here on, we work in Euclidean signature, although there is no obstruction to continuing back to Lorentzian signature. We would like to consider a static junction in these coordinates, meaning that our brane is located at $\rho_i = \rho_i^*$ in each spacetime patch. In each patch of spacetime $\mathcal{M}_i$ the coordinates $\rho_i$ will range from $-\infty < \rho < \rho_i^*$. The induced metric on the brane is therefore

$$\mathrm{d}^2\hat{s} \equiv h_{ab}\mathrm{d}\hat{x}^a\mathrm{d}\hat{x}^b = L_{\mathrm{I}}^2\cosh^2\left(\frac{\rho_{\mathrm{I}}^*}{L_{\mathrm{I}}}\right)\left(\frac{\mathrm{d}y_{\mathrm{I}}^2+\mathrm{d}\tau_{\mathrm{I}}^2}{y_{\mathrm{I}}^2}\right) = L_{\mathrm{II}}^2\cosh^2\left(\frac{\rho_{\mathrm{II}}^*}{L_{\mathrm{II}}}\right)\left(\frac{\mathrm{d}y_{\mathrm{II}}^2+\mathrm{d}\tau_{\mathrm{II}}^2}{y_{\mathrm{II}}^2}\right)\,, \quad (2.25)$$

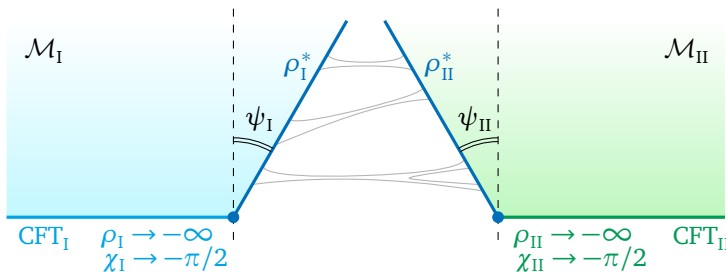

Figure 4: The definition of the coordinates $\psi_{\mathrm{I}}$ and $\psi_{\mathrm{II}}$ as the geometric angles between the brane and the normal direction of the conformal boundaries.

and the first junction condition enforces the relation

$$y_{\mathrm{I}} = y_{\mathrm{II}} , \qquad \tau_{\mathrm{I}} = \tau_{\mathrm{II}} , \qquad L_{\mathrm{I}} \cosh\left(\frac{\rho_{\mathrm{I}}^*}{L_{\mathrm{I}}}\right) = L_{\mathrm{II}} \cosh\left(\frac{\rho_{\mathrm{II}}^*}{L_{\mathrm{II}}}\right) . \tag{2.26}$$

The extrinsic curvatures can be readily computed

$$
\begin{aligned}
K_{\mathrm{I},ab} &= \left.\frac{1}{2}\frac{\partial h_{ab}}{\partial \rho_{\mathrm{I}}}\right|_{\rho_{\mathrm{I}}=\rho_{\mathrm{I}}^*} = \frac{1}{L_{\mathrm{I}}} \tanh\left(\frac{\rho_{\mathrm{I}}^*}{L_{\mathrm{I}}}\right) h_{ab} , \\
K_{\mathrm{II},ab} &= -\left.\frac{1}{2}\frac{\partial h_{ab}}{\partial \rho_{\mathrm{II}}}\right|_{\rho_{\mathrm{II}}=\rho_{\mathrm{II}}^*} = -\frac{1}{L_{\mathrm{II}}} \tanh\left(\frac{\rho_{\mathrm{II}}^*}{L_{\mathrm{II}}}\right) h_{ab} ,
\end{aligned}
\tag{2.27}
$$

where the sign difference comes from the fact that the outward normals on either side point in opposite directions. The second junction condition therefore takes the form

$$\frac{1}{L_{\mathrm{I}}} \tanh\left(\frac{\rho_{\mathrm{I}}^*}{L_{\mathrm{I}}}\right) + \frac{1}{L_{\mathrm{II}}} \tanh\left(\frac{\rho_{\mathrm{II}}^*}{L_{\mathrm{II}}}\right) = T . \tag{2.28}$$

These two conditions are solved by

$$\tanh\left(\frac{\rho_{\mathrm{I}}^*}{L_{\mathrm{I}}}\right) = \frac{L_{\mathrm{I}}}{2T}\left(T^2 + \frac{1}{L_{\mathrm{I}}^2} - \frac{1}{L_{\mathrm{II}}^2}\right) , \qquad \tanh\left(\frac{\rho_{\mathrm{II}}^*}{L_{\mathrm{II}}}\right) = \frac{L_{\mathrm{II}}}{2T}\left(T^2 + \frac{1}{L_{\mathrm{II}}^2} - \frac{1}{L_{\mathrm{I}}^2}\right) . \tag{2.29}$$

Since $-1 < \tanh(x) < 1$, one obtains a consistency constraint on the tension, namely

$$T_{\min} = \left|\frac{1}{L_{\mathrm{I}}} - \frac{1}{L_{\mathrm{II}}}\right| < T < \frac{1}{L_{\mathrm{I}}} + \frac{1}{L_{\mathrm{II}}} = T_{\max} . \tag{2.30}$$

In the following it will be useful to parametrize the location of the brane by its angle with respect to the perpendicular to the conformal boundaries in each patch. These angles are defined as follows:

$$\sin(\psi_{\mathrm{I},\mathrm{II}}) = \tanh\left(\frac{\rho_{\mathrm{I},\mathrm{II}}^*}{L_{\mathrm{I},\mathrm{II}}}\right) , \qquad \text{or} \qquad \psi_{\mathrm{I},\mathrm{II}} = \chi_{\mathrm{I},\mathrm{II}}^* , \tag{2.31}$$

and shown in Figure 4.

In later sections we will consider the *large tension limit*, meaning

$$T \to T_{\max} = \frac{1}{L_{\mathrm{I}}} + \frac{1}{L_{\mathrm{II}}} . \tag{2.32}$$

To understand the geometry in this regime it is useful write

$$T^2 = \frac{1}{L_{\text{I}}^2} + \frac{1}{L_{\text{II}}^2} + \frac{2 - \delta^2}{L_{\text{I}} L_{\text{II}}} \tag{2.33}$$

and expand for $\delta \to 0$. In this limit

$$\sin(\psi_{\text{I}}) = \tanh\left(\frac{\rho_{\text{I}}^*}{L_{\text{I}}}\right) = 1 - \frac{L_{\text{I}}^2}{2(L_{\text{I}} + L_{\text{II}})^2} \delta^2 + \mathcal{O}(\delta^4) \,,$$

$$\sin(\psi_{\text{II}}) = \tanh\left(\frac{\rho_{\text{II}}^*}{L_{\text{II}}}\right) = 1 - \frac{L_{\text{II}}^2}{2(L_{\text{I}} + L_{\text{II}})^2} \delta^2 + \mathcal{O}(\delta^4) \,. \tag{2.34}$$

Geometrically, this means that

$$\psi_{\text{I}} = \frac{\pi}{2} - \frac{L_{\text{I}}}{L_{\text{I}} + L_{\text{II}}} \delta + \mathcal{O}(\delta^2) \,, \qquad \psi_{\text{II}} = \frac{\pi}{2} - \frac{L_{\text{II}}}{L_{\text{I}} + L_{\text{II}}} \delta + \mathcal{O}(\delta^2) \,, \tag{2.35}$$

and thus in the large tension limit both $\text{AdS}_3^{\text{I}}$ and $\text{AdS}_3^{\text{II}}$ are reconstructed, since the brane approaches the conformal boundary. This fact will be important in section 4, in order to derive the island formula from the RT prescription.

## 2.6 The BCFT limit

It is sometimes useful to consider the limiting situation in which one of the two radii, let's say $L_{\text{II}}$ for concreteness, is much larger than the other. To gain intuition about this scenario, let us think of the limit $L_{\text{I}} \to 0$ first [51], although the latter lies well outside the semi-classical gravitational regime. If we just apply the Brown-Henneaux relation for $\text{AdS}_3/\text{CFT}_2$—eq. (2.2)—we find that the central charge of one of the two CFTs vanishes in this limit. Given that the central charge measures the number of degrees of freedom of a conformal field theory, the limit of $c_I \to 0$ eliminates all degrees of freedom of the $\text{CFT}_{\text{I}}$. No excitation can be transmitted from $\text{CFT}_{\text{II}}$ through the interface, which therefore acts as a boundary. In other words, the ICFT reduces to a BCFT. This argument can be made precise by using unitarity and its consequences on Cardy gluing conditions [35, 52]: the vanishing of the transmission coefficient is a necessary and sufficient condition for a conformal interface to be factorizing. We can now go back to the limit

$$\frac{L_{\text{I}}}{L_{\text{II}}} \to 0 \,, \tag{2.36}$$

which allow us to keep both $L_{\text{I}}$ and $L_{\text{II}}$ large in Planck units. It is intuitive, and also rigorously true due to unitarity [35, 52], that the transmission coefficient vanishes in the strict limit. Hence, eq. (2.36) corresponds to a BCFT limit which we can take without abandoning the classical regime.

Accordingly, one also expects the bulk brane to effectively become an EOW brane. We shall confirm this expectation in section 5, where the limit (2.36) will be performed on results found in previous sections for AdS/ICFT, and the BCFT expectations met. From this point of view, we would like to regard the AdS/BCFT framework as a subset of AdS/ICFT. The interest around this idea is that the bulk dual to an ICFT has the same topology as the asymptotically AdS spaces familiar from AdS/CFT. This allows us to easily extend some of the entries of the usual AdS/CFT dictionary. Taking the BCFT limit at the end, one derives the corresponding rules for AdS/BCFT, which are then justified and do not need to be separately conjectured. An example is the famous Takayanagi prescription to compute entanglement entropies in AdS/BCFT [17, 47], where the RT geodesics are allowed also to end on the EOW brane. Such a rule arises naturally from the interpolation CFT $\to$ ICFT $\to$ BCFT, as we will show in section 5.1.

It is interesting to note that, while the BCFT limit works perfectly in classical gravity, accounting for $1/c$ corrections might be subtle and require modifications. Indeed, since $L_{\mathrm{I}}/L_{\mathrm{II}} = c_{\mathrm{I}}/c_{\mathrm{II}}$, we cannot disregard transmission effects across the thin brane while keeping $1/c_{\mathrm{II}}$ contributions.

# 3 Correlation functions of heavy operators in ICFT

This section will provide a detailed review of the *geodesic approximation* [53] for bulk scalar field correlation functions (e.g. $\langle \phi(x)\phi(y) \rangle$) and boundary ICFT correlators (e.g. $\langle \mathcal{O}(x)\mathcal{O}(y) \rangle$). In particular, we derive below that, absent scalar field interactions localized at the brane between $\mathrm{AdS_I}$ and $\mathrm{AdS_{II}}$, the two-point function is computed by the geodesic in the large mass limit. Some of the relevant geodesics cross the brane: we point out that they are continuous and once differentiable in the coordinate system which makes the metric continuous, due to the Israel–Lanczos conditions. Upon taking the BCFT limit, smoothness of the geodesics will be responsible for Takayanagi's prescription for computing entanglement entropy in simple holographic BCFT setups [17].

## 3.1 Geodesic approximation

Primary operators $\mathcal{O}$ with $\Delta \gg 1$, are dual to scalar fields of mass $mL \approx \Delta$, meaning they are *heavy* in AdS units. The two point correlation function can thus be estimated using the geodesic approximation. In the holographic dual of ICFT described above (see figure 1), some geodesics will inevitably intersect the brane, especially if we are considering correlation functions such as $\langle \mathcal{O}_{\mathrm{I}}(x)\mathcal{O}_{\mathrm{II}}(y) \rangle$. We therefore need to understand the geodesic approximation applied to this case.

Let us begin by briefly reviewing the geodesic approximation in the absence of a brane. Consider a free scalar field $\phi$ of mass $m$, with Euclidean action

$$S[\phi] = \int \mathrm{d}^3 x \, \sqrt{g} \left( \frac{1}{2} g^{\mu\nu} \partial_\mu \phi \partial_\nu \phi + \frac{1}{2} m^2 \phi^2 \right) , \tag{3.1}$$

where $g_{\mu\nu}$ is the metric on (Euclidean) $\mathrm{AdS_3}$. The two point function is simply the propagator,

$$\langle \phi(x_1)\phi(x_2) \rangle = \int \mathcal{D}\phi(x) \, \phi(x_1)\phi(x_2) \, e^{-S[\phi]} , \tag{3.2}$$

which can be expressed in the worldline formalism as

$$\langle \phi(x_1)\phi(x_2) \rangle = \int\limits_{\substack{u(1)=x_2 \\ u(0)=x_1}} \frac{\mathcal{D}u(\tau)\mathcal{D}e(\tau)}{\mathrm{Vol(Gauge)}} \, e^{-S[u(\tau),e(\tau)]} \equiv \langle x_2 | x_1 \rangle , \tag{3.3}$$

(see e.g. [54,55]). In the equation above

$$S[u(\tau),e(\tau)] = \int_0^1 \mathrm{d}\tau \left( \frac{\dot{u}^2}{2e} + \frac{m^2 e}{2} \right) \tag{3.4}$$

is the worldline action,

$$\dot{u}^2 = g_{\mu\nu}(u)\dot{u}^\mu \dot{u}^\nu \tag{3.5}$$

and $e(\tau)$ is an *einbein* along the wordline. One way to see that this functional integral is designed to give us the propagator is to look at the constraint equation that arises from varying

$S$ with respect to $e(\tau)$:[5]

$$H \equiv g^{\mu\nu}(u)p_\mu p_\nu + m^2 = 0 \,, \qquad p_\mu \equiv i\frac{\partial L}{\partial \dot{u}^\mu} \,. \tag{3.6}$$

Upon canonically quantizing the theory, $H$ will be promoted to an operator, and the above constraint tells us that

$$\langle x_2|H|x_1\rangle = \hat{H}\langle x_2|x_1\rangle = 0 \,, \tag{3.7}$$

where $\hat{H}$ is a differential operator. Determining the form of $\hat{H}$ by canonically quantizing the classical expression (3.6) will generally suffer from ordering ambiguities due to the coordinate dependence in the background metric $g^{\mu\nu}$. Luckily, braver souls have attempted this before us [56,57]. Moreover we expect by general covariance that:

$$\hat{H} = -\Box_{x_{1,2}} + m^2 \,, \tag{3.8}$$

where $\Box$ is the Laplacian for the metric $g_{\mu\nu}$ and the coordinate it acts on depends on if it is taken to act to the right or to the left in (3.7). Taken together this means our worldline path integral is a generally covariant expression for a propagator from $x_1$ to $x_2$ on the background $g_{\mu\nu}$.[6]

Alternatively, we can integrate out the einbein entirely, leaving us with the standard Nambu-Goto action along the worldline, meaning our two-point function can be expressed as:

$$\langle \phi(x_1)\phi(x_2)\rangle = \int_{\substack{u(1)=x_2 \\ u(0)=x_1}} \mathcal{D}u(\tau)\, e^{-m\int_0^1 d\tau\sqrt{\dot{u}^2}} \,. \tag{3.9}$$

This latter expression admits a saddle point approximation in the limit of large mass:

$$\langle \phi(x_1)\phi(x_2)\rangle \sim \sum_{\mathcal{P}} e^{-m d_{\mathcal{P}}(x_1,x_2)} \,, \tag{3.10}$$

where $\mathcal{P}$ is the set of geodesic paths connecting $x_1$ and $x_2$ and $d_{\mathcal{P}}(x_1,x_2)$ is the length of the trajectory. To connect this calculation to that of a CFT two-point function for a primary operator of dimension $\Delta$, we identify

$$m^2 L^2 = \Delta(\Delta - 2) \approx \Delta^2 \tag{3.11}$$

and

$$\langle \mathcal{O}(x_1)\mathcal{O}(x_2)\rangle = \lim_{z_1, z_2 \to 0} \frac{1}{z_1^\Delta z_2^\Delta}\langle \phi(x_1)\phi(x_2)\rangle \,, \tag{3.12}$$

where we have used the notation $x_1 \equiv (z_1, \vec{x}_1)$ and similarly for $x_2$ in the Poincaré coordinates of (2.17).

## 3.2 Scalar field on a thin-brane background

The case we are interested in deals with a scalar field on a background that has a thin brane between $\mathcal{M}_\mathrm{I}$ and $\mathcal{M}_\mathrm{II}$. Let us denote the scalar field as $\phi_\mathrm{I}(x)$ if $x \in \mathcal{M}_\mathrm{I}$ and $\phi_\mathrm{II}(x)$ if $x \in \mathcal{M}_\mathrm{II}$. The Euclidean action for the scalar field can now be written as:

$$S[\phi] = \int_{\mathcal{M}_\mathrm{I}} d^3x\,\sqrt{g}\left(\frac{1}{2}g^{\mu\nu}\partial_\mu\phi_\mathrm{I}\partial_\nu\phi_\mathrm{I} + \frac{1}{2}m^2\phi_\mathrm{I}^2\right) + \int_{\mathcal{M}_\mathrm{II}} d^3x\,\sqrt{g}\left(\frac{1}{2}g^{\mu\nu}\partial_\mu\phi_\mathrm{II}\partial_\nu\phi_\mathrm{II} + \frac{1}{2}m^2\phi_\mathrm{II}^2\right) \,. \tag{3.13}$$

---

[5]The extra factor of $i$ in the definition of $p_\mu$ stems from our choice of a Euclidean target space.

[6]The correct delta function at coincident points is also accounted for by eq. (3.3), as can be seen for instance by going to a locally inertial frame when the two insertions are close, and comparing to the flat space version of the world-line path integral. The latter explicitly gives the expected $(p^2 + m^2)^{-1}$.

Varying the above action produces the following boundary term at the brane:

$$\delta S[\phi] = \int \mathrm{d}^2 y \, \sqrt{h} \left( \delta \phi_{\mathrm{I}} \partial_n \phi_{\mathrm{I}} - \delta \phi_{\mathrm{II}} \partial_n \phi_{\mathrm{II}} \right) , \tag{3.14}$$

where $y$ parameterizes the coordinates along the brane, $h_{\mu\nu}$ is the induced metric on the brane, and $n^\mu$ a unit normal to the brane (such that $\partial_n \equiv n^\mu \partial_{y^\mu}$). This contribution vanishes if we set, for example, decoupled Dirichlet or Neumann boundary conditions. However, we want to identify the scalar field across the brane, meaning we impose that the field configuration is continuous across the surface, thus we cannot independently vary $\phi_{\mathrm{I}}$ and $\phi_{\mathrm{II}}$ along the brane. Vanishing of (3.14) therefore requires that the normal derivatives also be equal. In formulas:

$$\phi_{\mathrm{I}}(y) = \phi_{\mathrm{II}}(y) , \qquad \partial_n \phi_{\mathrm{I}}(y) = \partial_n \phi_{\mathrm{II}}(y) . \tag{3.15}$$

With these boundary conditions we can reconsider the discussion above for the scalar field in AdS$_3$. Since the dynamics of a scalar field in each space is given by a second order (partial) differential equation, (3.15) implies that $\phi_{\mathrm{I}}(x)$ and $\phi_{\mathrm{II}}(x)$ can naturally be extended to a global scalar field $\phi(x)$ defined on the spacetime $\mathcal{M}_{\mathrm{I}} \cup \mathcal{M}_{\mathrm{II}}$, with:

$$\phi(x)\Big|_{x \in \mathrm{AdS}_{\mathrm{I}}} = \phi_{\mathrm{I}}(x) , \qquad \phi(x)\Big|_{x \in \mathrm{AdS}_{\mathrm{II}}} = \phi_{\mathrm{II}}(x) . \tag{3.16}$$

Moreover, based on the general arguments above, the propagator of this global field $\phi(x)$ can again be computed in the worldline formalism:

$$\langle \phi(x_1) \phi(x_2) \rangle \sim \sum_{\mathcal{P}'} e^{-m d_{\mathcal{P}'}(x_1, x_2)} , \tag{3.17}$$

where $\mathcal{P}'$ is a path in $\mathcal{M}_{\mathrm{I}} \cup \mathcal{M}_{\mathrm{II}}$ that can, in principle, cross the brane many times. Indeed, away from the brane, the same argument as in the previous subsection ensures that eq. (3.17) solves the Klein-Gordon equation. At the location of the brane, we only have to check the boundary conditions (3.15): we will show in the next subsection that geodesics across the interface precisely obey them. However, since the AdS scale jumps across the brane, we deduce that this scalar field is dual to operators of *different* conformal dimensions in the CFT$_{\mathrm{I,II}}$, namely:

$$m^2 L_{\mathrm{I,II}}^2 = \Delta_{\mathrm{I,II}} (\Delta_{\mathrm{I,II}} - 2) . \tag{3.18}$$

Taking this into account, the correlation function of, for instance, operators placed on either side of the boundary can be read off from the bulk formula using:

$$\langle \mathcal{O}_{\mathrm{I}}(x_1) \mathcal{O}_{\mathrm{II}}(x_2) \rangle = \lim_{z_1, z_2 \to 0} \frac{1}{z_1^{\Delta_{\mathrm{I}}} z_2^{\Delta_{\mathrm{II}}}} \langle \phi_{\mathrm{I}}(x_1) \phi_{\mathrm{II}}(x_2) \rangle . \tag{3.19}$$

Analogously to the rules familiar from the usual AdS/CFT dictionary, the setup can be generalised beyond the simple gluing condition (3.15). We could have considered adding scalar field self-interactions localized on the brane, as was considered in [58]. For instance, we could add to the effective action for the field $\phi$ a polynomial term of the form

$$\int_{\mathcal{S}} \mathrm{d}^2 y \, \sqrt{h} \, V(\phi, \nabla \phi) \subset S_{\mathrm{brane}} , \tag{3.20}$$

then the interactions present in $V(\phi)$ will affect the correlation functions. For example, a tadpole in $V(\phi)$ would generate non–trivial correlation functions between $\phi(x)$ on the conformal boundary and on the brane. Or similarly, if $V(\phi)$ includes a $\phi^n$ interaction, then one would need to include the appropriate $n$-point vertices on the brane and fix their positions by minimizing the length of the incident geodesics. We do not analyze these cases in this paper, especially because we will be ultimately interested in the application of the formalism to the computation of entanglement entropy, where the need for computing geodesics arises from the Ryu-Takayanagi prescription, rather than from a specific form of the scalar potential.

### 3.3 Continuity and smoothness of the geodesic crossing the brane

Let us try and understand the implications these considerations have when the geodesic approximation is valid. We will now show that the Israel–Lanczos conditions imply that the geodesics are continuous and once differentiable across the brane (and hence in the class $C^1$). As discussed in the previous subsection, these are the only saddles contributing in the worldline formalism, as long as eq. (3.15) holds, and in the absence of localized couplings along the brane.

Taking inspiration from [59], let us introduce a system of coordinates in a neighborhood of the interface $\mathcal{S}$ such that the local topology of spacetime is $\mathbb{R} \times \mathcal{S}$ in $\mathcal{M}_\mathrm{I} \cup \mathcal{M}_\mathrm{II}$. Specifically, this parametrization can be constructed using the set of geodesics labeled by a function $\lambda(x^\mu)$ such that the locus $\lambda = 0$ lies on the surface $\mathcal{S}$. Moreover, these geodesics will be constructed and such that their first derivative is normal to $\mathcal{S}$. Thus the function $\lambda(x^\mu)$, appropriately normalized, defines the proper distance (with sign) of the point $x^\mu$ from the surface $\mathcal{S}$. The unit normal to the brane, up to a normalization, is then

$$n_\alpha = \partial_\alpha \lambda \Big|_{\lambda=0} . \tag{3.21}$$

Using the Heaviside $\theta$–function, the metric on $\mathcal{M}_\mathrm{I} \cup \mathcal{M}_\mathrm{II}$ in the vicinity of $\mathcal{S}$ can then be conveniently written as

$$g_{\alpha\beta} = \theta(\lambda)g_{\alpha\beta}^\mathrm{I} + \theta(-\lambda)g_{\alpha\beta}^\mathrm{II} , \tag{3.22}$$

where we remind the reader that $g_{\alpha\beta}^\mathrm{I,II}$ are the metrics on either side of the brane. The derivative of the metric is then

$$\begin{aligned}
\partial_\gamma g_{\alpha\beta} &= \theta(\lambda)\partial_\gamma g_{\alpha\beta}^\mathrm{I} + \partial_\gamma \theta(\lambda)g_{\alpha\beta}^\mathrm{I} + \theta(-\lambda)\partial_\gamma g_{\alpha\beta}^\mathrm{II} + \partial_\gamma \theta(-\lambda)g_{\alpha\beta}^\mathrm{II} \\
&= \theta(\lambda)\partial_\gamma g_{\alpha\beta}^\mathrm{I} + \theta(-\lambda)\partial_\gamma g_{\alpha\beta}^\mathrm{II} + \delta(\lambda)(g_{\alpha\beta}^\mathrm{I} - g_{\alpha\beta}^\mathrm{II})n_\gamma .
\end{aligned} \tag{3.23}$$

The first Israel–Lanczos condition,

$$g_{\alpha\beta}^\mathrm{I} - g_{\alpha\beta}^\mathrm{II} \Big|_{\lambda=0} = 0 , \tag{3.24}$$

removes the $\delta$–discontinuity from (3.23), leaving only step–like ones. Consequently, the connection

$$\Gamma^\mu{}_{\nu\rho} = \frac{1}{2}g^{\mu\alpha}(\partial_\nu g_{\alpha\rho} + \partial_\rho g_{\alpha\nu} - \partial_\alpha g_{\nu\rho}), \tag{3.25}$$

is at most step-wise discontinuous, while the Riemann curvature

$$R^\mu{}_{\nu\rho\sigma} = \partial_\rho \Gamma^\mu{}_{\nu\sigma} - \partial_\sigma \Gamma^\mu{}_{\nu\rho} + \Gamma^\mu{}_{\rho\alpha}\Gamma^\alpha{}_{\nu\sigma} - \Gamma^\mu{}_{\sigma\alpha}\Gamma^\alpha{}_{\nu\rho}, \tag{3.26}$$

has a $\delta$–like discontinuity, required to satisfy the Einstein equations (since the stress–energy tensor has itself a $\delta$–discontinuity at the brane). However, we can deduce from the the geodesic equation,

$$\frac{\mathrm{d}^2 x^\mu}{\mathrm{d}\lambda^2} + \Gamma^\mu{}_{\nu\rho}\frac{\mathrm{d}x^\nu}{\mathrm{d}\lambda}\frac{\mathrm{d}x^\rho}{\mathrm{d}\lambda} = 0 , \tag{3.27}$$

that the absence of a $\delta$–like discontinuity in the connection implies that both the geodesic and its derivative must be continuous. The upshot of this discussion, as we will show, is that since boundary-anchored geodesics are continuous and smooth, they satisfy very simple geometric conditions. In the next section we will show how to solve the simple geometrical problems associated with finding various different boundary-anchored geodesics.

### 3.4 A panoply of geodesics

Let us begin with a few facts about geodesics in locally AdS spacetimes. In Poincaré coordinates, geodesics minimize the action functional

$$\int \mathrm{d}s \frac{L}{z(s)} \sqrt{\left(\frac{d\tau}{ds}\right)^2 + \left(\frac{dx}{ds}\right)^2 + \left(\frac{dz}{ds}\right)^2} \,, \tag{3.28}$$

and it is straightforward to show that constant-$\tau$ geodesics trace out circular arcs:

$$\left(x - \frac{\sigma_1 + \sigma_2}{2}\right)^2 + z^2 = \left(\frac{\sigma_1 - \sigma_2}{2}\right)^2 \,, \tag{3.29}$$

parametrized by their endpoints on the cutoff surface boundary at $x = \sigma_{1,2}$ and $z = 0$. Importantly these circles are centered along the conformal boundary at $z = 0$.

#### 3.4.1 Brane-crossing geodesics at equal-time

We now have everything in place to compute the geodesic distance between points on either side of the defect. For simplicity we will first take them to lie at equal time $\tau = \tau' = 0$, but will generalize to arbitrary times later. The distance of the two points from the interface on the conformal boundary is denoted as $\sigma_\mathrm{I}$ and $\sigma_\mathrm{II}$ respectively. A pictorial representation of such a geodesic is shown in figure 5. This case was considered before, see [60, 61].

Our task is to find a continuous and smooth geodesic that lands on the boundary at particular marked points. But we have just argued that geodesics in AdS$_3$ are circles centered along the conformal boundary. Therefore, finding the length of the geodesic in figure 5 is equivalent to finding two circular arcs that smoothly connect the boundary endpoints through the brane.

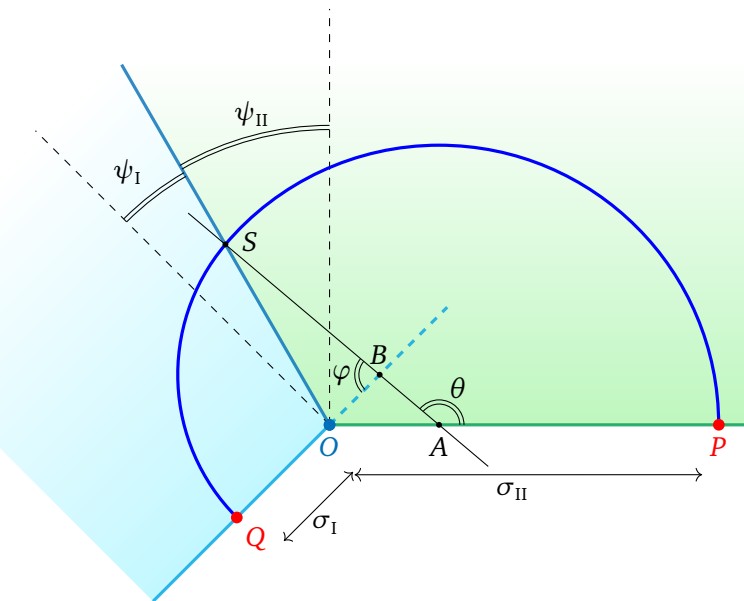

Figure 5: The smooth geodesic through the brane that connects two points on different CFTs which are respectively $\sigma_\mathrm{I}$ and $\sigma_\mathrm{II}$ away from the defect.

Let us denote by $A$ and $B$ the centers of the two circular arcs along their respective boundaries, as shown in figure 5. We will work this out specifically for the case where $\sigma_\mathrm{I} \leq \sigma_\mathrm{II}$, but

the result (3.48) is independent of this choice, as required by conformal invariance. Recall that these boundaries intersect at an angle along their common interface. The smoothness condition imposes that these circular arcs meet at a point $S$ on the brane that lies along a line that goes through both $A$ and $B$. To compute the length of the arcs we then need to compute the angles $\theta$ and $\varphi$ which subtend these arcs, and the radii of the circles $\overline{AP}$ and $\overline{BQ}$. First notice that

$$\widehat{AOB} = \psi_{\mathrm{I}} + \psi_{\mathrm{II}} \,. \tag{3.30}$$

Then, considering the angles of the triangle $\triangle AOB$ we get

$$\varphi = \pi + \psi_{\mathrm{I}} + \psi_{\mathrm{II}} - \theta \,. \tag{3.31}$$

The law of cosines applied to this triangle gives

$$\overline{OB}^2 = \overline{OA}^2 + \overline{AB}^2 + 2 \cdot \overline{OA} \cdot \overline{AB} \cdot \cos(\theta) \,, \tag{3.32}$$

while the law of sines leads to:

$$\overline{OB} \cdot \sin(\psi_{\mathrm{I}} + \psi_{\mathrm{II}}) = \overline{AB} \cdot \sin(\theta) \,. \tag{3.33}$$

On the other hand, looking at the triangle $\triangle OBS$, the law of sines yields

$$\overline{OS} \cos(\varphi - \psi_{\mathrm{I}}) = \overline{OB} \sin(\varphi) \,, \tag{3.34}$$

while the law of cosines applied to this triangle gives:

$$\overline{OS}^2 = \overline{OB}^2 + \overline{BS}^2 - 2 \cdot \overline{OB} \cdot \overline{BS} \cdot \cos(\varphi) \,. \tag{3.35}$$

We denote by $r$ the radius of the arc in $\mathrm{AdS_I}$ (blue region) and by $R$ the radius of the arc in the green $\mathrm{AdS_{II}}$ (green region). Thus:

$$\overline{AP} = \overline{AS} = R \,, \qquad \text{and} \qquad \overline{BQ} = \overline{BS} = r \,. \tag{3.36}$$

We also note the relations

$$\overline{OA} = \sigma_{\mathrm{II}} - R \,, \qquad \overline{OB} = r - \sigma_{\mathrm{I}} \,, \qquad \overline{AB} = R - r \,. \tag{3.37}$$

Recall that in our notation $\sigma_{\mathrm{I,II}}$ are positive quantities denoting distances along the boundary from the interface at $O$. Using (3.31) along with the two relations (3.33)-(3.34), we can write

$$R = r + (r - \sigma_{\mathrm{I}}) \sin(\psi_{\mathrm{I}} + \psi_{\mathrm{II}}) \csc(\theta) \,, \qquad \overline{OS} = (r - \sigma_{\mathrm{I}}) \sin(\psi_{\mathrm{I}} + \psi_{\mathrm{II}} - \theta) \sec(\theta - \psi_{\mathrm{II}}) \,. \tag{3.38}$$

Plugging these into the remaining two equations gives

$$(r - \sigma_{\mathrm{I}}) \left[ (r - \sigma_{\mathrm{I}}) \sin(\psi_{\mathrm{I}} + \psi_{\mathrm{II}}) + (r - \sigma_{\mathrm{II}}) \sin(\psi_{\mathrm{I}} + \psi_{\mathrm{II}}) \right] \sin(\psi_{\mathrm{I}} + \psi_{\mathrm{II}})$$
$$= (\sigma_{\mathrm{I}} + \sigma_{\mathrm{II}} - 2r)(\sigma_{\mathrm{I}} - \sigma_{\mathrm{II}}) \cos^2\left(\frac{\theta}{2}\right) \,, \tag{3.39}$$

and

$$r^2 + (r - \sigma_{\mathrm{I}}) \left[ (r - \sigma_{\mathrm{I}}) \cos(\psi_{\mathrm{I}}) \cos(2\theta - \psi_{\mathrm{I}} - 2\psi_{\mathrm{II}}) \sec^2(\theta - \psi_{\mathrm{II}}) + 2r \cos(\theta - \psi_{\mathrm{I}} - \psi_{\mathrm{II}}) \right] = 0 \,. \tag{3.40}$$

We now have two equations in the remaining two unknowns $(r, \theta)$. Being quadratic equations in $r$, it is important that we keep certain limits in mind when selecting the correct solution branch. Namely, when $\psi_{\mathrm{I}} = \psi_{\mathrm{II}} = 0$, we expect:

$$r = R = \frac{\sigma_{\mathrm{I}} + \sigma_{\mathrm{II}}}{2} \,, \qquad \cos(\theta) = \frac{\sigma_{\mathrm{I}} - \sigma_{\mathrm{II}}}{\sigma_{\mathrm{I}} + \sigma_{\mathrm{II}}} \,, \qquad \psi_{\mathrm{I}}, \psi_{\mathrm{II}} \to 0 \,. \tag{3.41}$$

The solutions to (3.39) and (3.40) that satisfy (3.41) are:

$$r = \frac{1}{2} \csc\left(\frac{\varphi}{2}\right) \sec\left(\frac{\psi_{\mathrm{I}} + \psi_{\mathrm{II}}}{2}\right) \left[ \sigma_{\mathrm{II}} \cos\left(\frac{\theta}{2}\right) - \sigma_{\mathrm{I}} \cos\left(\frac{\theta}{2} + \varphi\right) \right], \qquad (3.42)$$

$$R = \frac{1}{2} \csc\left(\frac{\theta}{2}\right) \sec\left(\frac{\psi_{\mathrm{I}} + \psi_{\mathrm{II}}}{2}\right) \left[ \sigma_{\mathrm{I}} \cos\left(\frac{\varphi}{2}\right) - \sigma_{\mathrm{II}} \cos\left(\frac{\varphi}{2} + \theta\right) \right]. \qquad (3.43)$$

Setting (3.42) equal to (3.43) (recalling that $\varphi = \pi + \psi_{\mathrm{I}} + \psi_{\mathrm{II}} - \theta$) along with a healthy amount of massaging gives us the following equation for $\theta$:

$$\sigma_{\mathrm{I}} \cos\left(\theta - \frac{\psi_{\mathrm{I}} + 3\psi_{\mathrm{II}}}{2}\right) + \sigma_{\mathrm{II}} \cos\left(\theta + \frac{\psi_{\mathrm{I}} - \psi_{\mathrm{II}}}{2}\right) = (\sigma_{\mathrm{I}} - \sigma_{\mathrm{II}}) \cos\left(\frac{\psi_{\mathrm{I}} - \psi_{\mathrm{II}}}{2}\right). \qquad (3.44)$$

Taking $\sigma_{\mathrm{I}} = \sigma_{\mathrm{II}}$, one sees that the solution is $\theta = \psi_{\mathrm{II}} + \frac{\pi}{2}$. More generally, the above equation can be rewritten as a quadratic equation in $\cos(\theta)$, with solution:

$$\cos(\theta) = \frac{\cos\left(\frac{\psi_{\mathrm{I}} - \psi_{\mathrm{II}}}{2}\right)}{\sigma_{\mathrm{I}}^2 + \sigma_{\mathrm{II}}^2 + 2\sigma_{\mathrm{I}}\sigma_{\mathrm{II}}\cos(\psi_{\mathrm{I}} + \psi_{\mathrm{II}})} \times$$
$$\left\{ -\sigma_{\mathrm{II}}^2 \cos\left(\frac{\psi_{\mathrm{I}} - \psi_{\mathrm{II}}}{2}\right) + \sigma_{\mathrm{I}}^2 \cos\left(\frac{\psi_{\mathrm{I}} + 3\psi_{\mathrm{II}}}{2}\right) + 2\sigma_{\mathrm{I}}\sigma_{\mathrm{II}}\sin(\psi_{\mathrm{II}})\sin\left(\frac{\psi_{\mathrm{I}} + \psi_{\mathrm{II}}}{2}\right) \right.$$
$$\left. - \left[ \sigma_{\mathrm{I}} \sin\left(\frac{\psi_{\mathrm{I}} + 3\psi_{\mathrm{II}}}{2}\right) - \sigma_{\mathrm{II}} \sin\left(\frac{\psi_{\mathrm{I}} - \psi_{\mathrm{II}}}{2}\right) \right] \right\} \times$$
$$\times \sqrt{\left[ \frac{(\sigma_{\mathrm{I}} + \sigma_{\mathrm{II}})^2 - (\sigma_{\mathrm{I}} - \sigma_{\mathrm{II}})^2 \cos(\psi_{\mathrm{I}} - \psi_{\mathrm{II}}) + 4\sigma_{\mathrm{I}}\sigma_{\mathrm{II}}\cos(\psi_{\mathrm{I}} + \psi_{\mathrm{II}})}{2\cos^2\left(\frac{\psi_{\mathrm{I}} - \psi_{\mathrm{II}}}{2}\right)} \right]}, \qquad (3.45)$$

and the branch of the square root in (3.45) is selected such that $\cos\theta = -\sin\psi_{\mathrm{II}}$ in the limit $\sigma_{\mathrm{I}} = \sigma_{\mathrm{II}}$.

To convert this data into a geodesic length, recall that, as reviewed in section 2.4, any two points $x^\mu$ and $x'^\mu$ in AdS$_3$ can be labeled by their embedding coordinates on the hyperboloid: $X^A$ and $X'^A$ where $A = 0, \ldots 3$. The geodesic distance $d$ between any two points is therefore determined by

$$-G_{\mu\nu}X^\mu X'^\nu = L^2 \cosh\left(\frac{d}{L}\right). \qquad (3.46)$$

In the Poincaré coordinates of (2.17), we thus have

$$d = L \cosh^{-1}\left( \frac{(\tau - \tau')^2 + (x - x')^2 + z^2 + z'^2}{2z\,z'} \right). \qquad (3.47)$$

For the section of the geodesic in AdS$_{\mathrm{I}}$, we can treat the point $B$ as the origin of our coordinates. Thus we want to compute the distance between a point at $(x, z) = (-r, \varepsilon_{\mathrm{I}})$ and the point $(x', z') = (-r\cos\varphi, r\sin\varphi)$, while in the AdS$_{\mathrm{II}}$ we want to compute the distance between a point at $(x, z) = (R, \varepsilon_{\mathrm{II}})$ and the point $(x', z') = (R\cos\theta, R\sin\theta)$, both with $\tau = \tau'$. In this expression we are allowing for the distinct possibility that the CFT duals have different UV cutoffs. The leading order result as $\varepsilon_{\mathrm{I,II}} \to 0$ is:

$$d(\sigma_{\mathrm{I}}, \sigma_{\mathrm{II}}) = L_{\mathrm{I}} \log\left[ \frac{2r}{\varepsilon_{\mathrm{I}}} \tan\left(\frac{\varphi}{2}\right) \right] + L_{\mathrm{II}} \log\left[ \frac{2R}{\varepsilon_{\mathrm{II}}} \tan\left(\frac{\theta}{2}\right) \right], \qquad (3.48)$$

where we remind the reader of the relation (3.31) between $\theta$ and $\varphi$. Notice that when $\sigma_{\mathrm{II}} = \sigma_{\mathrm{I}} = \sigma$, the geodesic length simplifies to

$$R = r = \sigma, \qquad \varphi = \psi_{\mathrm{I}} + \frac{\pi}{2}, \qquad \theta = \psi_{\mathrm{II}} + \frac{\pi}{2}, \qquad (3.49)$$

so that

$$d(\sigma, \sigma) = L_{\mathrm{I}} \log\left[\frac{2\sigma}{\varepsilon_{\mathrm{I}}} \tan\left(\frac{\psi_{\mathrm{I}}}{2} + \frac{\pi}{4}\right)\right] + L_{\mathrm{II}} \log\left[\frac{2\sigma}{\varepsilon_{\mathrm{II}}} \tan\left(\frac{\psi_{\mathrm{II}}}{2} + \frac{\pi}{4}\right)\right]. \tag{3.50}$$

This is expected, since in this case $A$ and $B$ meet at the origin $O$ in figure 5. Using (2.31), we can rewrite this as:

$$d(\sigma, \sigma) = \rho_{\mathrm{I}}^{*} + \rho_{\mathrm{II}}^{*} + L_{\mathrm{I}} \log\left[\frac{2\sigma}{\varepsilon_{\mathrm{I}}}\right] + L_{\mathrm{II}} \log\left[\frac{2\sigma}{\varepsilon_{\mathrm{II}}}\right]. \tag{3.51}$$

The dramatic simplification of this formula is easy to understand from the point of view of the dual field theory. Recall that the exponential of the distance computes a correlation function in the ICFT. When the points are in mirroring positions with respect of the interface, like in eq. (3.51), the conformal group acts on the correlator in the same way as on a one-point function in a BCFT. This can be easily seen, for instance, via the folding trick—see *e.g.* [36]. In turn, the dependence on the coordinates of a one-point function is completely fixed by symmetry, and this yields eq. (3.51).

### 3.4.2 Points on different sides at generic positions and conformal properties

The result of the previous section allows us also to compute the geodesic length between two points with $\tau_1 \neq \tau_2$. Indeed, we can take advantage of the invariance of the geodesic distance under the isometries. Our main interest lies in the dual CFT, so let us think about the points on the conformal boundary. It is easy to show [62,63] that knowledge of a two-point function on the line $\tau_1 = \tau_2$ is sufficient to reconstruct the correlator everywhere. Indeed, there is only one cross ratio for two points with a flat boundary:

$$\xi = \frac{(\sigma_{\mathrm{I}} - \sigma_{\mathrm{II}})^2 + (\tau_{\mathrm{I}} - \tau_{\mathrm{II}})^2}{4\sigma_{\mathrm{I}}\sigma_{\mathrm{II}}}. \tag{3.52}$$

$\xi$ is positive and vanishes when the operators are in the mirroring position discussed above, and diverges as one point is brought close to the interface. Specifically, the two-point function of our scalar primary must take the form

$$\langle \mathcal{O}(x_{\mathrm{I}}) \mathcal{O}(x_{\mathrm{II}}) \rangle = \frac{1}{\sigma_{\mathrm{I}}^{\Delta_{\mathrm{I}}} \sigma_{\mathrm{II}}^{\Delta_{\mathrm{II}}}} \tilde{g}(\xi), \tag{3.53}$$

with $\tilde{g}$ a function which is not fixed by symmetry. Comparing this equation with eq. (3.17) and with the extrapolate dictionary (3.19), we see that the geodesic distance (3.48) must take the form

$$d(\sigma_{\mathrm{I}}, \sigma_{\mathrm{II}}) = L_1 \log\frac{\sigma_{\mathrm{I}}}{\varepsilon_{\mathrm{I}}} + L_2 \log\frac{\sigma_{\mathrm{II}}}{\varepsilon_{\mathrm{II}}} + g(\xi), \tag{3.54}$$

where

$$g(\xi) = L_{\mathrm{I}} \log\left[\frac{2r}{\sigma_{\mathrm{I}}} \tan\left(\frac{\varphi}{2}\right)\right] + L_{\mathrm{II}} \log\left[\frac{2R}{\sigma_{\mathrm{II}}} \tan\left(\frac{\theta}{2}\right)\right]. \tag{3.55}$$

Consistently, $g(\xi)$ only depends on the ratio $\sigma_{\mathrm{I}}/\sigma_{\mathrm{II}}$. Eq. (3.54) encodes the dependence of the distance, and therefore of the correlator, from generic positions of the endpoints.[7] Indeed, one can invert eq. (3.52) in the $\tau_{\mathrm{I}} = \tau_{\mathrm{II}}$ case, and obtain

$$\frac{\sigma_{\mathrm{I}}}{\sigma_{\mathrm{II}}} = 1 + 2\xi \pm 2\sqrt{\xi(\xi+1)}. \tag{3.56}$$

---

[7]We are keeping the cutoffs $\epsilon_{\mathrm{I}}$ and $\epsilon_{\mathrm{II}}$ fixed. This is the correct procedure to obtain physical correlators in the CFT. Of course, if one was to apply an AdS isometry to the endpoints of the geodesic, including their Poincaré coordinate distance from the conformal boundary, the length would stay the same.

Plugging eq. (3.56) into eq. (3.55), one gets the explicit expression as a function of the cross ratio. It's a nice check of eq. (3.48) that the result does not depend on the branch chosen for the square root. Indeed, swapping branches corresponds to sending $\sigma_I/\sigma_{II} \to \sigma_{II}/\sigma_I$. This is a conformal transformation (for instance, an inversion), and is the only invariance which is not explicit in eq. (3.48). We checked this fact numerically: it would be nice to simplify eq. (3.48) further to write it as a simple function of $\xi$. Finally, the correlator (3.53) can be computed in any configuration by simply evaluating $\xi$ in eq. (3.52) at the desired position.

For completeness, we report the explicit form of a conformal Killing vector which produces the generic configuration from the $\tau_I = \tau_{II}$ case. The subgroup of the isometries of AdS which preserves the position of the brane is simply the one which fixes $\rho$ in the parametrization (2.15). This is the isometry group of $AdS_2$, which acts on the complex coordinate $w = \tau + i y$ via the $sl(2,\mathbb{R})$ transformation

$$w \to \frac{aw+b}{cw+d}, \quad ad-bc \neq 0, \quad a,b,c,d \in \mathbb{R}. \tag{3.57}$$

If $c \neq 0$, the transformation includes a special conformal transformation on each slice (including the conformal boundary). It is easy to check that this isometry will do the job. For instance, the choice $a = d = 1$, $b = -c = \lambda$ generates circular orbits in the $(\tau, y)$ plane, as a function of $\lambda$, which leave invariant the point $(0, 1)$, as well as the the boundary $y = 0$.[8] Clearly, placing one operator at $\sigma_I = 1$ and varying the position of the other on the $\tau = 0$ line, we obtain any configuration, up to a translation and a dilatation. One can easily map the transformation (3.57) to Poincaré coordinates, if needed, via eq. (2.19). For instance, the special conformal $a = d = 1$, $b = 0$ $c \in \mathbb{R}$ becomes

$$x \to \frac{x}{1+2c\tau+c^2(\tau^2+x^2+z^2)}, \quad z \to \frac{z}{1+2c\tau+c^2(\tau^2+x^2+z^2)}, \quad \tau \to \frac{\tau+c(\tau^2+x^2+z^2)}{1+2c\tau+c^2(\tau^2+x^2+z^2)}, \tag{3.58}$$

which is nothing but a special conformal transformation for the full $AdS_3$ with parameter along the $\tau$ direction.

### 3.4.3 Points on the same side at $\tau = 0$

In this section we will be interested in CFT two-point functions with operator insertions restricted to one side of the interface. Recall that we have chosen to parametrize our CFTs such that $c_I < c_{II}$, without loss of generality. This has implications for correlation functions of operators placed in the $CFT_{II}$ region, such as $\langle \mathcal{O}_{II}(x_1)\mathcal{O}_{II}(x_2)\rangle$. This is because, when $L_I < L_{II}$, there exist boundary-anchored geodesics such as the one depicted in figure 6, that probe the geometry behind the brane. However no such geodesic exists for e.g. $\langle \mathcal{O}_I(x_1)\mathcal{O}_I(x_2)\rangle$ when $L_I < L_{II}$.

Proving the existence of such geodesics follows the same logic as before: given points $P$ and $Q$, we want to show that there exist continuous and smooth geodesics that can be built piecewise out of circular arcs centered on the AdS boundary (or its continuation through the brane). We will instead ask a related question: for each (allowable) choice of the point $C$ and the angle $\alpha$ defined in figure 6, how many give geodesics that end on the pair of points $P$ and $Q$?

Considering figure 6, by simple geometric reasoning we note that (recall that $\widehat{AOB} = \psi_I + \psi_{II}$)

$$\widehat{BSO} = \gamma - \psi_{II} - \frac{\pi}{2}, \tag{3.59}$$

---

[8]The action of this conformal Killing vector is just a rotation of the half sphere obtained as a stereographic projection of the upper half plane

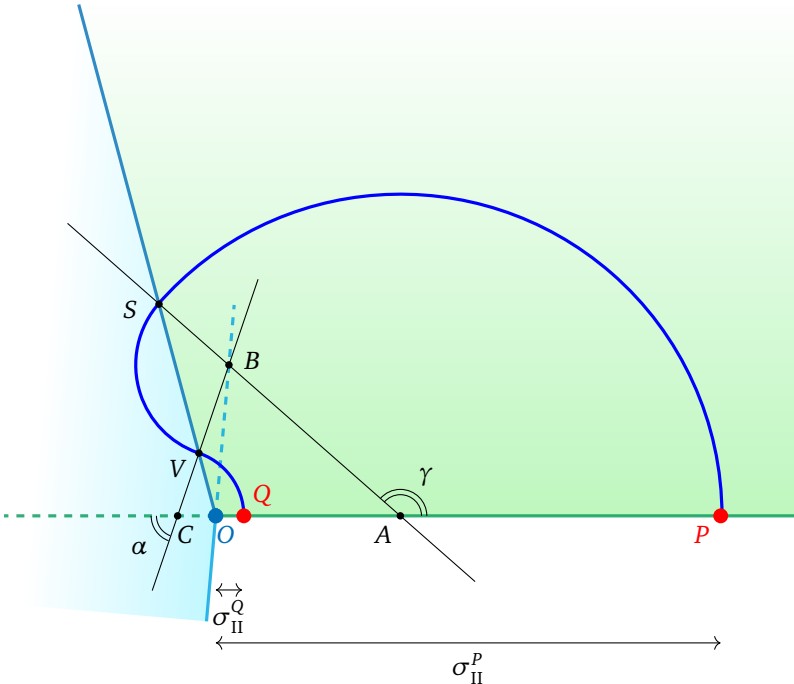

**Figure 6:** Geometric construction to find geodesics that explore the space behind the brane. The point $O$ refers to the origin where the defects lies, not one of the boundary points of the geodesic.

while

$$\widehat{BVS} = \frac{\pi}{2} - \alpha + \psi_{\text{II}} \ . \tag{3.60}$$

Since the points $S$ and $V$ lie along a circular arc in the AdS$_{\text{I}}$ region, the triangle $\triangle BSV$ is isosceles and therefore $\widehat{BSO} = \widehat{BVS}$, implying that the angles $\alpha$ and $\gamma$ are related:

$$\gamma = \pi + 2\psi_{\text{II}} - \alpha \ . \tag{3.61}$$

Moreover, defining:

$$\overline{OQ} = \sigma_{\text{II}}^Q \ , \qquad \overline{OP} = \sigma_{\text{II}}^P \ , \qquad \overline{OC} = \lambda \ , \tag{3.62}$$

we note that the point $A$ is entirely determined in terms of $\lambda$ and $\alpha$:

$$\overline{OA} = \frac{\lambda \sin(\alpha) \sin(\alpha + \psi_{\text{I}} - \psi_{\text{II}})}{\sin(\alpha - 2\psi_{\text{II}}) \sin(\psi_{\text{I}} + \psi_{\text{II}} - \alpha)} \ , \tag{3.63}$$

and, using elementary geometry, we readily compute the location of the endpoints $P$ and $Q$ along the boundary in terms of $\lambda$ and $\alpha$ as well:

$$\sigma_{\text{II}}^Q = \lambda \left( \frac{\cos(\psi_{\text{II}})}{\cos(\alpha - \psi_{\text{II}})} - 1 \right) \ , \qquad \sigma_{\text{II}}^P = \lambda \frac{\sin(\alpha) \sin(\alpha + \psi_{\text{I}} - \psi_{\text{II}})[\cos(\psi_{\text{II}}) + \cos(\alpha - \psi_{\text{II}})]}{\sin(\alpha - 2\psi_{\text{II}}) \sin(\psi_{\text{I}} + \psi_{\text{II}} - \alpha) \cos(\alpha - \psi_{\text{II}})} \ . \tag{3.64}$$

Given the external data $\{\sigma_{\text{II}}^Q, \sigma_{\text{II}}^P\}$, we would like to determine the allowed set of $\lambda$ and $\alpha$ that satsify (3.64). Going forward, it will be convenient to parametrize $\alpha$ as :

$$\alpha = \psi_{\text{II}} + \cos^{-1} \left[ \frac{\cos(\psi_{\text{II}})}{1+s} \right] \ , \tag{3.65}$$

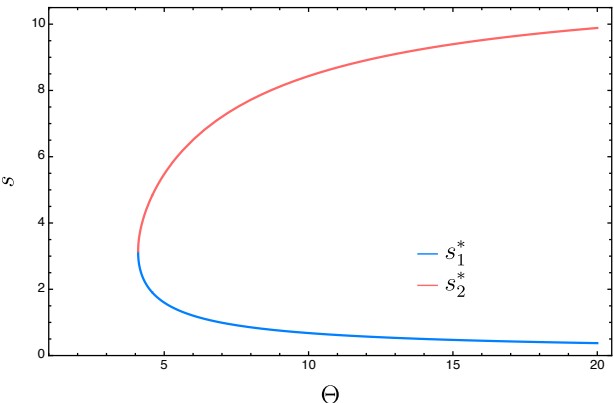

Figure 7: The solutions $s^*_{1,2}$ for $\psi_{\mathrm{I}} = 1.5$ and $\psi_{\mathrm{II}} = 0.5$. Recall that $\Theta \equiv \frac{\sigma^P_{\mathrm{II}}}{\sigma^Q_{\mathrm{II}}} \geq 1$. For $\Theta$ large enough there are always two geodesics that connects them of the kind of figure 6.

for $s > -1 + \cos(\psi_{\mathrm{II}})$. The geometric interpretation of $s$ is clear plugging this parametrization into the first equation in (3.64), which gives $\lambda = \sigma^Q_{\mathrm{II}}/s$. Notice that $\lambda > 0$ must hold for $\psi_{\mathrm{II}} \in [0, \pi/2]$, thus our parameter $s$ is valued in:

$$
s > \begin{cases} 0\,, & \psi_{\mathrm{II}} \in [0, \pi/2]\,, \\ -1 + \cos(\psi_{\mathrm{II}})\,, & \psi_{\mathrm{II}} \in [-\pi/2, 0]\,. \end{cases} \tag{3.66}
$$

Substituting this into the second equation in (3.64) we obtain

$$
\Theta \equiv \frac{\sigma^P_{\mathrm{II}}}{\sigma^Q_{\mathrm{II}}} = -\frac{(s+2)}{s} \frac{(1 + 2s(s+2)) \cos(\psi_{\mathrm{I}}) - \cos(\psi_{\mathrm{I}} + 2\psi_{\mathrm{II}}) + 2 \sin(\psi_{\mathrm{I}} + \psi_{\mathrm{II}}) \sqrt{(1+s)^2 - \cos^2(\psi_{\mathrm{II}})}}{(1 + 2s(s+2)) \cos(\psi_{\mathrm{I}}) - \cos(\psi_{\mathrm{I}} + 2\psi_{\mathrm{II}}) - 2 \sin(\psi_{\mathrm{I}} + \psi_{\mathrm{II}}) \sqrt{(1+s)^2 - \cos^2(\psi_{\mathrm{II}})}}\,. \tag{3.67}
$$

In figure 7, we plot a contour in the $\Theta - s$ plane that satisfies (3.67) for given $\psi_{\mathrm{I,II}}$. We note that for $\Theta$ sufficiently large, there are two branches of solutions, which we denote by by $s^*_1$ and $s^*_2$, meaning that there exists either two geodesics of the type depicted in figure 6 for a given set of endpoints, or none.

Since we have taken $L_{\mathrm{I}} < L_{\mathrm{II}}$ we expect paths of the type in figure 6 to exist by virtue of trying to take a shortcut through the spacetime that is more highly curved. Interestingly, no solutions to (3.67) exist if we take $\psi_{\mathrm{II}} > \psi_{\mathrm{I}}$, meaning that the above analysis above actually shows that no such geodesics exist for $L_{\mathrm{I}} > L_{\mathrm{II}}$.

The geodesics found in this section also have a natural interpretation in the BCFT limit of our setup, which we explore in Section 5.3. Recall that this limit consists of taking $L_{\mathrm{I}}/L_{\mathrm{II}} \to 0$, thus these saddles should, in principle, exist. Indeed, in holographic BCFT setups one often finds disconnected geodesics (as well as reflecting ones) which contribute to correlation functions, as has been noted in [58]. In Section 5.3 we reinterpret such connected and disconnected contributions as a limit of the ICFT geodesics that cross the brane. The reader may consult Appendix A for a computation of the length of these geodesics.

**Saddles with more brane crossings:** We have only touched on the possibility of geodesics that cross the brane at most twice. One may also consider paths that cross the brane more times. However, their existence is not general, and they may appear only for specific regions of the parameter space. Moreover, the length of these geodesics is always bigger than the saddles found above. We can prove this formally. Let $I$ be the set of points in which the path $\mathcal{P}$ crosses the brane. If the cardinality card$(I) \geq 3$, there is always a pair of points in $I$ that

are connected by an arc that lies in the AdS with smaller curvature. However, there always is another geodesic stretching between the same two points and lying in the other AdS. The new path $\mathcal{P}'$, obtained by performing this replacement, is shorter.[9] This means that such paths do not contribute to the computation of entanglement entropies or correlation functions at leading order.

# 4 Entanglement entropy and the island formula

In this section, we compute the entanglement entropies of various subsystems of the baths depicted in figure 2, and their time evolutions according to different choices of the Hamiltonian. We shall analyze in detail the cases of two semi-infinite subsystems in the vacuum—subsection 4.1—and in the thermofield double state—subsection 4.2.

By repeating the computation in the intermediate perspective of figure 1, we will show that the island formula of [6,64] *coincides* with the Ryu–Takayanagi prescription, in the limit when the tension is large and gravity on the brane is weakly coupled. This idea is not new [9,14], but the present context allows to exhibit it quantitatively. In particular, rather than just finding agreement between the two sides in the leading cutoff dependent term, we find a detailed match, including cases where the dependence on the position of the entangling surface is not fixed by symmetry. Furthermore, the RT and the island formulas agree at the level of universal, physical quantities. Our main example is the Page time, which is independent of the infinite entanglement entropy of the QFT vacuum.

At the technical level, we perform the computations by putting to good use the formalism developed in section 3. While for the correlation functions the geodesic approximation relies on the operators being heavy, the RT prescription in $AdS_3$ precisely instructs us to compute geodesics homologous to the entangling surface, which consists of two points.[10] Let us also emphasize that the standard RT prescription is invoked here, where both endpoints of the geodesic are on the boundary. This is a nice conceptual gift of the ICFT setup: the island formula follows from a rigorously proven tool [65], rather than from its well established by still conjectural extension [17] to BCFT. Nevertheless, in section 5 we shall partly bridge this gap by recovering the BCFT rule as a limit of the ICFT one.

## 4.1 Semi-infinite intervals in the vacuum

Let us start by asking the following question. What is the von Neumann entropy of the density matrix obtained from the vacuum state of the system in figure 2 by tracing over the conformal interface and a part of the two baths? Specifically, we choose the entangling surface to be composed of two points at distances $\sigma_{\mathrm{I}}$ and $\sigma_{\mathrm{II}}$ from the defect respectively, as depicted on the left panel of figure 8.

The answer in the large $c$ limit is given by the RT formula [66]:

$$S_{\sigma_{\mathrm{I}},\sigma_{\mathrm{II}}} = \frac{d(\sigma_{\mathrm{I}},\sigma_{\mathrm{II}})}{4G_{(3)}} \,, \tag{4.1}$$

where $d(\sigma_{\mathrm{I}},\sigma_{\mathrm{II}})$ is the length of the geodesic depicted in figure 8. Using eq. (3.48), one finds

$$S_{\sigma_{\mathrm{I}},\sigma_{\mathrm{II}}} = \frac{c_{\mathrm{I}}}{6} \log\left[\frac{2r}{\varepsilon}\tan\left(\frac{\varphi}{2}\right)\right] + \frac{c_{\mathrm{II}}}{6}\log\left[\frac{2R}{\varepsilon}\tan\left(\frac{\theta}{2}\right)\right] \,, \tag{4.2}$$

---

[9]Although in general $\mathcal{P}'$ is not an extremizer because it's not $C^1$.
[10]See also the discussion at the end of subsection 5.1.



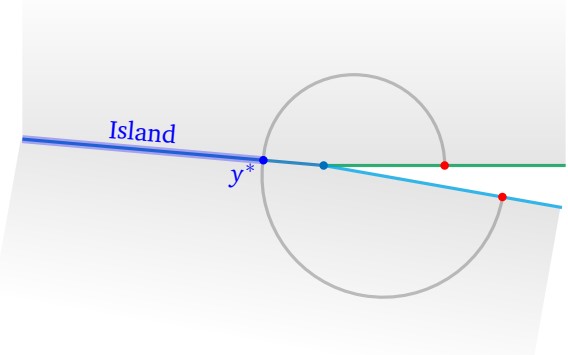
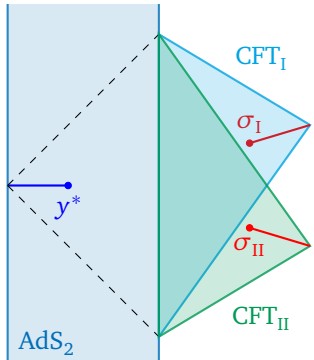

Figure 8: In the spirit of double holography we can interpret the geodesic of Figure 5 in a a two dimensional perspective. In particular we can integrate out the bulk degrees of freedom and consider the induced effective action on the brane. Then, the bulk geodesic can be interpreted as the island in the intermediate two dimensional picture. In particular, for large values of the tension, one can prove that the point where the geodesic of Figure 5 crosses the brane is the position where the island appears minimizing the $2d$ generalized entropy.

where the dependence on $\sigma_{\mathrm{I}}$ and $\sigma_{\mathrm{II}}$ is through $r$, $R$, $\theta$ and $\varphi$. The explicit expressions can be found in eqs. (3.31,3.42,3.43,3.45), while the geometric meanings of these parameters are clear from figure 5, where $r = BQ$ and $R = AP$.

Eq. (4.2) is complicated in general, but it becomes transparent if we take $\sigma_{\mathrm{I}} = \sigma_{\mathrm{II}} = \sigma$. In this case, as remarked in subsection 3.4, conformal symmetry is more constraining, and forces the $\sigma$ dependence to simplify—see eq. (3.50):

$$S_\sigma = \frac{c_{\mathrm{I}}}{6} \log \frac{2\sigma}{\varepsilon} + \frac{c_{\mathrm{II}}}{6} \log \frac{2\sigma}{\varepsilon} + \frac{\rho_{\mathrm{I}}^* + \rho_{\mathrm{II}}^*}{4G_{(3)}} \ . \tag{4.3}$$

Subtracting the bulk contribution off, we read the interface entropy

$$S_{\mathrm{int}} = \frac{\rho_{\mathrm{I}}^* + \rho_{\mathrm{II}}^*}{4G_{(3)}} \ . \tag{4.4}$$

Let us now reinterpret eq. (4.2) from the point of view of the 'brane+baths' system. As advocated in the introduction, we should expect a simple picture to emerge in the large tension limit of eq. (2.32):

$$T \to T_{\max} \ . \tag{4.5}$$

The brane is pushed towards the boundary—see eq. (2.35)—and the isometries of AdS$_3$ act as conformal transformations on points of the brane. In other words, the theory on the brane is now a CFT coupled to a (weakly) fluctuating metric, deformed by a UV cutoff which behaves locally like the standard cutoff of Poincaré AdS. The nature of the CFT on the brane also follows from the geometry. Integrating out the bulk on either side, we end up, by symmetry, with two copies of the Polyakov action with central charges $c_{\mathrm{I}}$ and $c_{\mathrm{II}}$. All in all, we have two holographic CFTs, each defined on the whole real line. On half of the line, the CFTs are decoupled and live on a frozen conformally flat metric. On the other half, the CFTs still interact through the common metric, as it is clear from the fact that the transmission coefficient does not vanish even when the tension is the maximal one [42]. It would be interesting to compute energy transmission and reflection in the 'brane+baths' picture.

With this in mind, it is easy to realize what the geodesic in the left panel of figure 8 computes. The arc on either side is an RT surface for $CFT_I$ and $CFT_{II}$ respectively. It computes the entanglement entropy of two intervals stretching between $\sigma_{I/II}$ respectively and $y^*$. Also, recall that the value $y^*$ can be obtained by minimizing the length of all the curves which join $\sigma_I$ to $\sigma_{II}$, are piecewise geodesics, and pass through a point $y$ on the brane. These entropies can be computed via the universal CFT formula [67]. However, it pays off to be careful with the role of the cutoff. At the points $\sigma_{I/II}$, we should of course maintain the uniform cutoff $\varepsilon$ used so far. On the brane, the cutoff, chosen again as the coordinate distance from the boundary of AdS in Poincaré coordinates, is point dependent, and easily seen to be $\varepsilon_{\text{brane}}(y) = y \cos \psi$.[11] Explicitly, we expect to find

$$S_{\sigma_I \sigma_{II}} \sim \min_y \left[ \frac{c_I}{6} \log\left( \frac{(y+\sigma_I)^2}{y\varepsilon} \frac{1}{\cos(\psi_I)} \right) + \frac{c_{II}}{6} \log\left( \frac{(y+\sigma_{II})^2}{y\varepsilon} \frac{1}{\cos(\psi_{II})} \right) \right] \equiv S_{\text{island}} , \quad T \to T_{\text{max}} . \quad (4.6)$$

This is, remarkably, the island formula [6, 9, 68–70]. Indeed, the quantity to minimize is a special case of the generalized entropy:

$$S_{\text{gen}} = \frac{A(\partial I)}{4G_N} + S_{\text{vN}}(I \cup R) , \qquad S_{\text{island}} = \min_y S_{\text{gen}} , \quad (4.7)$$

where $I$ is the island and $R$ the region of the bath whose entanglement entropy we want to compute. In the present case, the island is the region highlighted in blue in figure 8, and the area term is missing because there is no Hilbert-Einstein term, nor a dilaton, on the brane.

Eq. (4.6) can be verified explicitly, also making precise how many terms in an expansion in $(T \to T_{\text{max}})$ can be matched. The equation

$$\partial_y S_{\text{gen}} = 0 \qquad \Rightarrow \qquad \frac{2c_I}{y+\sigma_I} - \frac{c_I}{y} + \frac{2c_{II}}{y+\sigma_{II}} - \frac{c_{II}}{y} = 0 , \quad (4.8)$$

implies that the minimum, *i.e.* the position of the quantum extremal surface (QES), is at

$$y^* = \frac{(c_I - c_{II})(\sigma_I - \sigma_{II}) + \sqrt{(c_I - c_{II})^2(\sigma_I - \sigma_{II})^2 + 4\sigma_I \sigma_{II}(c_{II} + c_{II})^2}}{2(c_I + c_{II})} . \quad (4.9)$$

Notice that the second solution of the quadratic related to (4.8) is always negative and $y^*$ must be positive, so we disregard it. In the limit $T \to T_{\text{max}}$, it can be checked that the position of the island $y^*$ approaches the point where the geodesic in the bulk crosses the brane. Moreover, the equality (4.6) is verified. Ideed, writing the tension of the brane as in eq. (2.33):

$$T^2 = \frac{1}{L_I^2} + \frac{1}{L_{II}^2} + \frac{2-\delta^2}{L_I L_{II}} , \quad (4.10)$$

in the $\delta \to 0$ limit one finds

$$\cos(\psi_I) = \frac{L_I}{L_I + L_{II}} \delta + \mathcal{O}(\delta^2) , \qquad \cos(\psi_{II}) = \frac{L_{II}}{L_I + L_{II}} \delta + \mathcal{O}(\delta^2) , \quad (4.11)$$

and the minimum of the generalized entropy can be expanded as

$$S_{\text{island}} = -\frac{c_I + c_{II}}{6} \log(\delta) + \frac{c_I}{6} \log\left[ \frac{(y^*+\sigma_I)^2}{y^*\varepsilon} \left( \frac{L_I + L_{II}}{L_I} \right) \right] + \frac{c_{II}}{6} \log\left[ \frac{(y^*+\sigma_{II})^2}{y^*\varepsilon} \left( \frac{L_I + L_{II}}{L_{II}} \right) \right] + \mathcal{O}(\delta) . \quad (4.12)$$

The 3d bulk computation (4.2) agrees with this result up to terms which vanish as $\delta \to 0$. This agreement is quite remarkable—we have matched a nontrivial formula (not fixed by symmetry) in ICFT using the QES prescription in the 2d braneworld, giving strong evidence of the validity of this formula in the setting of 2d gravity.

---

[11]Alternatively, one can think of the theory on the brane to live in $AdS_2$, and keep into account the appropriate Weyl factor, but, for consistency, no additional cutoff.

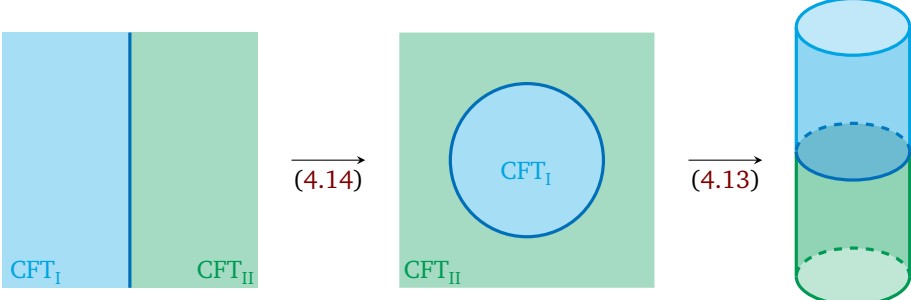

Figure 9: Series of transformations which map the ICFT from the plane to the cylinder. The thermofield double state is obtained by cutting open the Euclidean path integral in the last picture along a vertical plane passing through the origin.

## 4.2 Black hole evaporation in double holography and AdS/ICFT

The same ideas can be applied in the context of the black hole information paradox in which the radiation is collected in semi–infinite intervals [6, 9, 68–70], or in finite subregions [71]. The simplest setup consists of the thermofield double state, which is connected to the vacuum by a conformal transformation, as it is shown in figure 9. This approach was originally presented in [10] in the case of AdS/BCFT, using the RT prescription. Here we will extend this computation to the case of AdS/ICFT,[12] and we will complement it with the two dimensional computation in the intermediate picture. Again, this will allow us to derive the island formula from the RT prescription. The agreement between the two perspectives, in this case, is all the more relevant, because it implies that a UV finite quantity, the Page time, can be matched exactly.

Let us review the basic idea. On the conformal boundary, the thermofield double state is prepared by the Euclidean path integral on half of the infinite cylinder, with the defect running along the circle—see figure 10. The initial condition for the Lorentzian evolution consists therefore of *two* copies of the system in figure 2, whose state is entangled. Tracing over one of the copies, we get a thermal state for the other. The holographic dual to this state contains an eternal black string in $AdS_3$ [73], which crosses the brane and induces a horizon on it as well. The Lorentzian geometry is sketched in figure 11.

It is important to emphasize that Lorentzian time evolution on the two boundaries is obtained by Wick rotating the generator of the rotation on the circle, as appropriate for a finite temperature system. In figure 11, the corresponding Killing vector moves time 'upward' in the left asymptotic region and 'downward' in the right one.

The simplest quantum-information theoretic quantity to compute in this system is the entanglement entropy of the density matrix obtained tracing over both copies of the quantum dot. In the 'brane+bath' perspective, this is the entropy of the Hawking radiation deposited in the baths by the $AdS_2$ black hole. In this case we have four twist operators and not two, since we have doubled the system: they are marked with red dots in figures 10, 11 and 12. For simplicity, we choose them to lie all at the same distance from the quantum dots.[13] The time evolution we are interested in evolves 'upward' in both copies of the system. This is not a symmetry, and the entanglement entropy is time dependent. In particular, since the quantum

---

[12]In the recent paper [72] a related construction is used. In that case, the compact spatial direction in the two copies of the bath forces the two CFTs to be the same.

[13]The symmetries of the problem are then the same as in the BCFT case treated in [10], and the 3*d* computation is expected to match. Furthermore, the comparison to the island formula was presented for this case in [15]. However, using formulas from section 3, one can extend the result to the asymmetric case. In particular, the island phase is technically identical to the computation presented in subsection 4.1. Hence, the match to the island formula is guaranteed in the most general case.

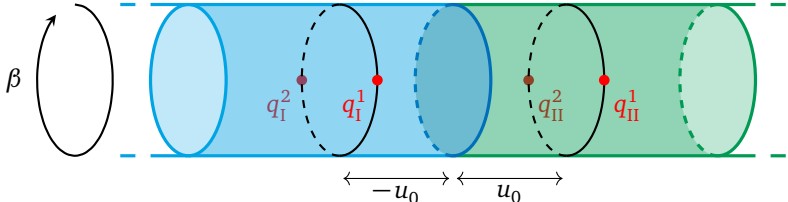

Figure 10: Thermofield double state of the system considered. The red dots are the twist operators. To obtain the Lorentzian evolution, one has to cut the cylinder at $v_E = 0$ and $v_E = \frac{i\beta}{2}$, and then perform the Wick rotation $v_E = iv$.

dot has finite thermodynamic entropy, the radiation cannot increase the entropy in the baths indefinitely.

As promised, we will compute such entropy in two ways: both using RT surfaces in three dimensions and using the island prescription in two dimensions, as it is sketched in figure 11. As in the previous section, we expect the two entropies to match in the limit of large tension, and that an island in two dimensions forms when the two RT saddles exchange dominance (the Page time $t_P$). Moreover, the points where the blue geodesics in figure 11 cross the brane are the boundaries of the island in two dimensions.

It is also useful to conformally map the CFTs from the cylinder to flat space. The Euclidean path integral now is done with the defect placed on a circle of length $l$, which is an arbitrary scale. Later it will only appear in a dimensionless combination with the UV cutoff $\varepsilon$, and cancel out from universal quantities like the Page time. The transformation is

$$p = l\, e^{\frac{2\pi}{\beta}q} \,, \tag{4.13}$$

where $p = \tilde{x} + i\tilde{\tau}$ is the complex coordinate on the plane and $q = u + iv_E$ is the complex coordinate on the cylinder, such that $v_E \simeq v_E + \beta$, as depicted in figure 10. Later we will consider only the Lorentzian evolution obtained by the Wick rotation $v_E = iv$. Correlation functions in the planar geometry can be computed by conformally mapping the circle to a straight line and using the results of section 3. The analytic continuation to real time described above now turns the defect into a hyperbola, and stationary observers in the thermofield double state lie on hyperbolic trajectories on the plane. This coordinate system is depicted in the left panel of figure 12. In details, we can connect the geometry of the previous section (coordinates $x$ and $t$) with the one of this section (coordinates $\tilde{x}$, $\tilde{t}$ in Lorentzian) with the AdS isometry which acts on the boundary by turning the flat interface into a circle. Explicitly,

$$\tilde{x}_i = \frac{x_i - \dfrac{x_i^2 + z_i^2 - t_i^2}{2l}}{1 - \dfrac{x_i}{l} + \dfrac{x_i^2 + z_i^2 - t_i^2}{4l^2}} + l \,,$$

$$\tilde{z}_i = \frac{z_i}{1 - \dfrac{x_i}{l} + \dfrac{x_i^2 + z_i^2 - t_i^2}{4l^2}} \,, \qquad \tilde{t}_i = \frac{t_i}{1 - \dfrac{x_i}{l} + \dfrac{x_i^2 + z_i^2 - t_i^2}{4l^2}} \,. \tag{4.14}$$

Notice that we are already in Lorentzian signature. In these coordinates the brane is the locus of points

$$\tilde{x}_I^2 - \tilde{t}_I^2 + (\tilde{z}_I - l\tan(\psi_I))^2 = \frac{l^2}{\cos^2(\psi_I)} \,, \tag{4.15}$$

on side I of the brane, and

$$\tilde{x}_{\text{II}}^2 - \tilde{t}_{\text{II}}^2 + (\tilde{z}_{\text{II}} + l\tan(\psi_{\text{II}}))^2 = \frac{l^2}{\cos^2(\psi_{\text{II}})} \,, \tag{4.16}$$

on side II. The brane ends on the conformal boundary along the hyperbola depicted in figure 12. As announced above, we take the twist operators at the same distance $-u_{\text{I}}^{1,2} = u_{\text{II}}^{1,2} = u_0$ in the cylinder geometry of figure 10. Their Lorentzian trajectories in the planar geometry are

$$\tilde{x}_{\text{I}}(q_{\text{I}}^1) = l e^{-\frac{2\pi}{\beta}u_0}\cosh\left(\frac{2\pi}{\beta}v\right), \qquad \tilde{t}_{\text{I}}(q_{\text{I}}^1) = l e^{-\frac{2\pi}{\beta}u_0}\sinh\left(\frac{2\pi}{\beta}v\right), \tag{4.17}$$

$$\tilde{x}_{\text{II}}(q_{\text{II}}^1) = l e^{\frac{2\pi}{\beta}u_0}\cosh\left(\frac{2\pi}{\beta}v\right), \qquad \tilde{t}_{\text{II}}(q_{\text{II}}^1) = l e^{\frac{2\pi}{\beta}u_0}\sinh\left(\frac{2\pi}{\beta}v\right), \tag{4.18}$$

$$\tilde{x}_{\text{I}}(q_{\text{I}}^2) = -l e^{-\frac{2\pi}{\beta}u_0}\cosh\left(\frac{2\pi}{\beta}v\right), \qquad \tilde{t}_{\text{I}}(q_{\text{I}}^2) = l e^{-\frac{2\pi}{\beta}u_0}\sinh\left(\frac{2\pi}{\beta}v\right), \tag{4.19}$$

$$\tilde{x}_{\text{II}}(q_{\text{II}}^2) = -l e^{\frac{2\pi}{\beta}u_0}\cosh\left(\frac{2\pi}{\beta}v\right), \qquad \tilde{t}_{\text{II}}(q_{\text{II}}^2) = l e^{\frac{2\pi}{\beta}u_0}\sinh\left(\frac{2\pi}{\beta}v\right). \tag{4.20}$$

The length of the orange geodesics in figure 11 and 12 grows with $v$ as

$$S_{\text{early}}^{\text{plane}} = \frac{c_{\text{I}}}{3}\log\left[\frac{2l}{\varepsilon}e^{-\frac{2\pi}{\beta}u_0}\cosh\left(\frac{2\pi}{\beta}v\right)\right] + \frac{c_{\text{II}}}{3}\log\left[\frac{2l}{\varepsilon}e^{\frac{2\pi}{\beta}u_0}\cosh\left(\frac{2\pi}{\beta}v\right)\right] \sim \frac{c_{\text{I}}+c_{\text{II}}}{3}\frac{2\pi}{\beta}v \,, \tag{4.21}$$

thus linearly in time after a short transient. To express this result in the coordinates of the cylinder, we have to take into account that the transformation (4.13) generates a Weyl factor between the metric on the plane and on the cylinder, namely

$$\mathrm{d}s_{\text{plane}}^2 = \mathrm{d}p\,\mathrm{d}\bar{p} = l^2\left(\frac{2\pi}{\beta}\right)^2 e^{\frac{2\pi}{\beta}(q+\bar{q})}\mathrm{d}q\,\mathrm{d}\bar{q} = \Omega^2(q,\bar{q})\,\mathrm{d}s_{\text{cyl}}^2 \,. \tag{4.22}$$

Taking this into account, the entropy in the cylinder geometry reads

$$S_{\text{early}}^{\text{cyl}} = \frac{c_{\text{I}}+c_{\text{II}}}{3}\log\left[\frac{\beta}{\pi\varepsilon}\cosh\left(\frac{2\pi}{\beta}v\right)\right] \sim \frac{c_{\text{I}}+c_{\text{II}}}{3}\frac{2\pi}{\beta}v \,, \tag{4.23}$$

This saddle dominates at early time, and does not cross the brane: eq. (4.23) is independent of the tension $T$. In terms of the conformal block decomposition of the correlator of twist operators, it is obtained by only keeping the identity operator in the fusion of each pair on either side of the interface. Notice moreover that the dependence on the position of the twist operators (namely $u_0$) drops from the final form. This is due to the fact that this saddle is insensitive to the presence of the defect, which is the only source of breaking of the translational symmetry of the cylinder.

On the other hand, the island saddle can be obtained by appropriately transforming the one obtained in the vacuum in the previous subsection. Applying the transformation (4.14) on eq. (4.3), accounting for the second pair of twist operators and appropriately transforming the cutoffs, we get for the entropy of the blue geodesics

$$S_{\text{late}}^{\text{plane}} = \frac{c_{\text{I}}}{3}\log\left[\frac{l}{\varepsilon}\frac{e^{\frac{4\pi}{\beta}u_0}-1}{e^{\frac{4\pi}{\beta}u_0}}\right] + \frac{c_{\text{II}}}{3}\log\left[\frac{l}{\varepsilon}\left(e^{\frac{4\pi}{\beta}u_0}-1\right)\right] + \frac{\rho_{\text{I}}^* + \rho_{\text{II}}^*}{2G_{(3)}} \,. \tag{4.24}$$

Taking into account the same Weyl transformation (4.22), we get

$$S_{\text{late}}^{\text{cyl}} = \frac{c_{\text{I}}+c_{\text{II}}}{3}\log\left[\frac{\beta}{\pi\varepsilon}\sinh\left(\frac{2\pi}{\beta}u_0\right)\right] + \frac{\rho_{\text{I}}^* + \rho_{\text{II}}^*}{2G_{(3)}} \,. \tag{4.25}$$

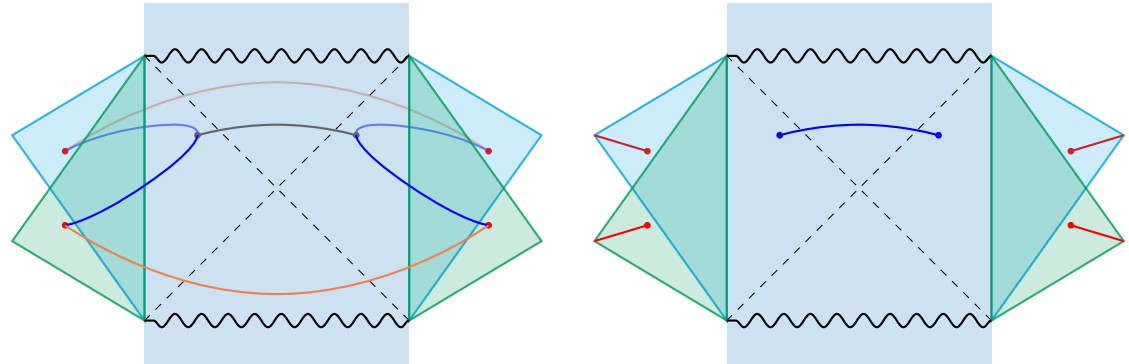

Figure 11: The information paradox for the thermofield double state. *Left*: the computation of the entanglement entropy of the radiation region using RT surfaces. The orange lines are the RT saddles that dominate at early times, in which the entropy increases linearly in time. The blue lines are the RT saddles that cross the brane, and are constant in time, thus saturate the entanglement entropy. The gray line is the island. *Right*: the same computation in the two dimensional perspective using the generalized entropy formula. The blue dots represent the QES, and the blue line is the island.

This saddle is constant in $v$ and dominates at late times. The result depends on the boundary entropy and on the position of the twist operators, as expected. Moreover, since we are considering a doubled system, the saturation is close to twice the interface entropy, provided we place the twist operators sufficiently close to the defect. In this case the interface entropy can thus be interpreted as twice the black hole entropy.

The Page curve is shown in Figure 12 (right), in which we can distinguish an early rising phase and a saturation phase. The exchange of dominances is at the Page time, defined as

$$S_{\text{early}}^{\text{cyl}}(v_P) = S_{\text{late}}^{\text{cyl}} , \qquad (4.26)$$

and this is a renormalization scheme independent quantity, as it can be checked by the fact that the cutoff $\varepsilon$ drops out when equating eqs. (4.23) and (4.25). Explicitly we find

$$v_P = \frac{\beta}{2\pi} \cosh^{-1}\left[ \sinh\left(\frac{2\pi}{\beta} u_0\right) \exp\left(\frac{6 S_{\text{int}}}{c_{\text{I}} + c_{\text{II}}}\right) \right] , \qquad (4.27)$$

where we remind the reader that

$$\frac{6 S_{\text{int}}}{c_{\text{I}} + c_{\text{II}}} = \frac{\rho_{\text{I}}^* + \rho_{\text{II}}^*}{L_{\text{I}} + L_{\text{II}}} . \qquad (4.28)$$

In the limit of large boundary degrees of freedom, $S_{\text{int}} \gg c_{\text{I}} + c_{\text{II}}$, for a fixed $u_0$ this becomes

$$v_P = \frac{\beta}{2\pi} \cosh^{-1}\left[ \sinh\left(\frac{2\pi}{\beta} u_0\right) \right] + \frac{3\beta}{\pi} \frac{S_{\text{int}}}{c_{\text{I}} + c_{\text{II}}} , \qquad (4.29)$$

in which we notice that the page time depends linearly on the ratio between $S_{\text{int}}$ and $c_{\text{I}} + c_{\text{II}}$. Thus in this limit, which is the one in where we will be able to match the 2D picture, the Page time is long with respect to the parameter $\beta$. Moreover, in the limit where $u_0 \gtrsim \beta$ this further simplify into

$$v_P = u_0 + \frac{3\beta}{\pi} \frac{S_{\text{int}}}{c_{\text{I}} + c_{\text{II}}} . \qquad (4.30)$$

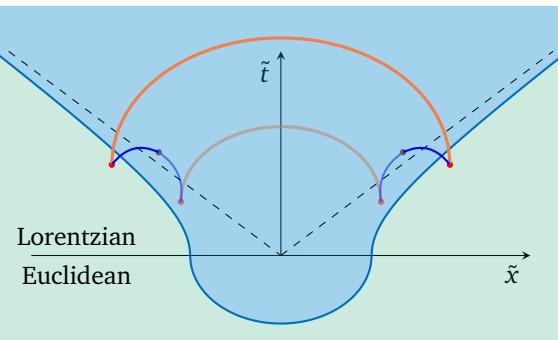
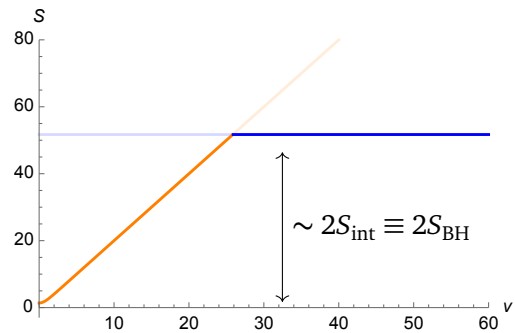

Figure 12: *Left*: the system in the planar thermofield double state (in Lorentzian signature). The brane (dark blue) becomes an hyperboloid, and the red dots during the dynamics follow hyperbolic trajectories. The orange RT surfaces are dominant at early times, and the entropy increases linearly. The blue RT surfaces are dominant at late times, and the entropy is constant. Such geodesics are the corresponding island saddle in the two dimensional perspective. *Right*: the dynamics of the entanglement entropies in time corresponding to the different RT surfaces (orange and blue), in the cylinder coordinates. The total entanglement entropy is always the smaller between (4.23) and (4.25), thus it increases (approximately) linearly until it saturates at (approximately) twice of the interface entropy. In this plot we have chosen $c_I = 6$, $c_{II} = 12$, $\beta = 2\pi$, $u_0 = 1$, $\varepsilon = 1$ and $\frac{\rho_I^* + \rho_{II}^*}{2G_{(3)}} = 50$.

The linear dependence on $u_0$, as emphasized in [10], is due to the *flying time* it takes the radiation to reach the twist operators.

The same computation can be done in the two dimensional perspective—see the right panel of figure 11. The reasoning is exactly the same as in subsection 4.1. In particular, since the early time contribution is only sensitive to the degrees of freedom in the baths, it automatically matches in the two perspectives, so we don't have to compute it again. On the other hand, for the island contribution we first notice that in the static coordinate system at $t = 0$, the twist operators $q^1_{I,II}$ are at position

$$x_I(q^1_I) = -2l \tanh\left(\frac{\pi u_0}{\beta}\right) , \qquad x_{II}(q^1_{II}) = 2l \tanh\left(\frac{\pi u_0}{\beta}\right) . \tag{4.31}$$

Then, the generalized entropy we must minimize in the static coordinate system is (for only one pair of twist)

$$S_{\text{gen}}^{\text{static}} = \frac{c_I}{6} \log\left[ \frac{(\tilde{y} + 2l \tanh(\pi u_0/\beta))^2}{\tilde{y}\,\varepsilon} \frac{1}{\cos(\psi_I)} \right] + \frac{c_{II}}{6} \log\left[ \frac{(\tilde{y} + 2l \tanh(\pi u_0/\beta))^2}{\tilde{y}\,\varepsilon} \frac{1}{\cos(\psi_{II})} \right] . \tag{4.32}$$

In the planar $p$-coordinates, accounting for both pairs and transforming as usual the cutoffs, this becomes

$$\begin{aligned} S_{\text{gen}}^{\text{plane}} = \frac{c_I}{3} \log &\left[ \frac{(\tilde{y} + 2l \tanh(\pi u_0/\beta))^2}{\tilde{y}\,\varepsilon} \frac{\left(e^{\frac{2\pi}{\beta} u_0} + 1\right)^2}{4 e^{\frac{4\pi}{\beta} u_0} \cos(\psi_I)} \right] \\ &+ \frac{c_{II}}{3} \log\left[ \frac{(\tilde{y} + 2l \tanh(\pi u_0/\beta))^2}{\tilde{y}\,\varepsilon} \frac{\left(e^{\frac{2\pi}{\beta} u_0} + 1\right)^2}{4 \cos(\psi_{II})} \right] , \end{aligned} \tag{4.33}$$

and finally on the cylinder

$$S_{\text{gen}}^{\text{cyl}} = \frac{c_{\text{I}} + c_{\text{II}}}{3} \log \left[ \left( \frac{\beta}{2\pi l} \right) \frac{(\tilde{y} + 2l \tanh(\pi u_0/\beta))^2}{\tilde{y}\,\varepsilon} \frac{\left( e^{\frac{2\pi}{\beta} u_0} + 1 \right)^2}{4 e^{\frac{2\pi}{\beta} u_0} \cos(\psi_{\text{I}})} \right] . \tag{4.34}$$

The minimum of (4.34) is at

$$\tilde{y}^* = 2l \tanh \left( \frac{\pi u_0}{\beta} \right) , \tag{4.35}$$

and plugging this into (4.34), we obtain

$$S_{\text{gen}}^{\text{cyl}} = \frac{c_{\text{I}} + c_{\text{II}}}{3} \log \left[ \frac{\beta}{\pi\varepsilon} \sinh \left( \frac{2\pi}{\beta} u_0 \right) \right] + \frac{c_{\text{I}}}{3} \log \left[ \frac{2}{\cos(\psi_{\text{I}})} \right] + \frac{c_{\text{II}}}{3} \log \left[ \frac{2}{\cos(\psi_{\text{II}})} \right] . \tag{4.36}$$

It is not hard to see that, in the large tension limit, the two-dimensional entropy (4.36) agrees with its three-dimensional counterpart (4.25), recalling the relation (2.31). While this result is only one conformal transformation away from the computation of subsection 4.1, it has a different relevance in this context. Indeed, the agreement obtained both at early and late times implies that the Page time matches in the two description. This is a well-defined quantity, free from any subtleties related to the choice of UV cutoffs. In the CFT, it can be expressed in terms of CFT data: the central charges and the defect entropy. It is remarkable that the island formula reproduces it exactly.

# 5 Seafaring from AdS/ICFT to AdS/BCFT

In this section, we take advantage of the BCFT limit defined in subsection 2.6 to show that virtually all the observables associated to the end-of-the-world (EOW) brane described in [17] can be obtained as limits of specific ICFT counterparts. We illustrate this fact by reproducing the geodesic approximation for correlators in a BCFT [17,58], as well as proving the prescription for computing the entanglement entropy of a semi-infinite interval [17].

## 5.1 Geodesics and entanglement entropy in BCFT from ICFT

To obtain the physics of BCFT from ICFT, we must tune the AdS lengths such that one becomes much smaller than the other, i.e.

$$\nu \equiv \frac{L_{\text{I}}}{L_{\text{II}}} \to 0 . \tag{5.1}$$

This ensures that $c_{\text{II}} \gg c_{\text{I}}$, meaning that the interface becomes impermeable to the transmission of excitations, as the CFT$_{\text{I}}$ has comparatively too few degrees of freedom per unit volume to acommodate a generic excitation coming in from the right.

How do we extract the vacuum geometry in this limit? First recall that the tension $T$ is constrained to lie between $T_{\min}$ and $T_{\max}$ as defined in (2.30). We can parametrize the range of possible $T$ via

$$T^2 = \frac{1}{L_{\text{I}}^2} + \frac{1}{L_{\text{II}}^2} + \frac{2\eta}{L_{\text{I}} L_{\text{II}}} , \tag{5.2}$$

where the constraint is now translated into $-1 < \eta < 1$. Let us rewrite (2.29), in terms of of these parameters:

$$\sin(\psi_{\text{I}}) = \frac{1 + \eta\,\nu}{\sqrt{1 + 2\eta\,\nu + \nu^2}} , \qquad \sin(\psi_{\text{II}}) = \frac{\eta + \nu}{\sqrt{1 + 2\eta\,\nu + \nu^2}} . \tag{5.3}$$

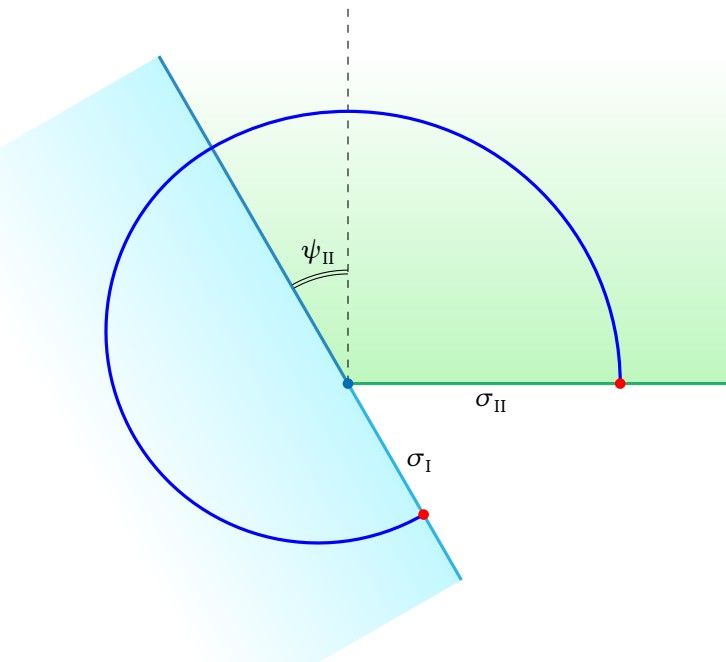

Figure 13: The geodesic for the two point function in the BCFT limit proposed in (2.36). In such regime the angle $\psi_I$ goes to $\pi/2$, the length doesn't depend on $\sigma_I$ anymore and the geodesic approaches the brane at a right angle, as proposed by Takayanagi in [17].

In the limit $\nu \to 0$, this means

$$\sin(\psi_I) \to 1 \,, \qquad \sin(\psi_{II}) = \eta \,. \tag{5.4}$$

Meaning that $\psi_I \to \pi/2$ and the geometry becomes that of figure 13, where we have also displayed a geodesic between two boundary points across the interface, as studied in section 3.4.1. Eq. (5.4) can be used to relate $\eta$ to the tension of the EOW brane.

Let us return to the types of geodesics shown in figure 13. We see clearly that in the limit $\nu \to 0$, the angle $\varphi$ as shown in figure 5 is tending to $\pi$, meaning that, according to (3.31),

$$\theta \to \frac{\pi}{2} + \psi_{II} \,. \tag{5.5}$$

For this configuration, $r$ and $R$ given in (3.42) and (3.43) tend to $\frac{\sigma_I + \sigma_{II}}{2}$ and $\sigma_{II}$ respectively. We can obtain the length of the geodesic connecting these two points by taking $L_I = \nu L_{II}$ in (3.48), yielding

$$d(\sigma_I, \sigma_{II}) \underset{\nu \to 0}{=} L_{II} \log\left[ \frac{2\sigma_{II}}{\varepsilon_{II}} \tan\left( \frac{\psi_{II}}{2} + \frac{\pi}{4} \right) \right] = \rho_{II}^* + L_{II} \log\left[ \frac{2\sigma_{II}}{\varepsilon_{II}} \right] \,, \tag{5.6}$$

where the dependence on $\sigma_I$ has dropped out entirely, and we have thus reproduced the result in [17].

Some interpretation is in order. A geodesic connecting two boundary points is usually associated with a two-point function. However, in sending $\nu \to 0$ we are sending the ratio of dimensions $\Delta_I/\Delta_{II}$ to zero as well, effectively producing the correlation function of $\mathcal{O}_{II}$ with the identity. Thus in the $\nu \to 0$ limit, the ICFT two point function effectively becomes a BCFT one point function, whose universal form is

$$\langle \mathcal{O}_{II}(\sigma_{II}) \rangle = \frac{a_{\mathcal{O}}}{(2\sigma_{II})^{\Delta_{II}}} \,, \tag{5.7}$$

as can be seen by exponentiating (5.6) using the geodesic approximation. In [17], this result was obtained by minimizing over the length of all curves ending on the EOW brane. That procedure yields a geodesic which arrives on the brane at a right angle. In figure 13, we see how this condition arises in the $\nu \to 0$ limit of the smooth geodesics considered in the ICFT construction: the geodesic in the $\mathrm{AdS_I}$ region becomes a complete semi-circle.

What about entanglement entropies? RT surfaces that compute EEs are codimension-2, so in the specific case of $\mathrm{AdS_3/CFT_2}$, they would correspond to geodesics, but not so in higher dimensions. Furthermore, the emergence of minimal surfaces in both three and higher dimensions does not stem from the worldline path integral of a massive particle, so we need an alternative argument to convince us that we should expect minimal surfaces to compute entanglement entropies in ICFT, and that their BCFT limits give rise to Takayanagi's prescription. Luckily the reasoning of [65] still holds in our holographic interface setup (even generalized to higher dimensions) since the crux of their argument relied on the $\mathbb{Z}_n$ symmetry of the $n-$replicated entangling region at the locally-asymptotically-AdS boundary—which one must consider when calculating the entanglement entropy using the standard replica trick. In this construction, the minimality of the RT surface stems from the leading order effects of the codimension-2 locus in the bulk left invariant under the action of this $\mathbb{Z}_n$, even as we take $n \to 1$. At the asymptotic boundary in ICFT we will still specify boundary conditions at the entangling surface, and will therefore have a $\mathbb{Z}_n$ symmetry in the replicated geometry, even if the entangling surface crosses the interface. Since the bulk metric is continuous due to the Israel-Lanczos conditions, there will again be a $\mathbb{Z}_n$-invariant codimension-2 surface that will ultimately be responsible for the minimal RT surfaces in ICFT whose effect will survive the $n \to 1$ limit. Having established that minimal RT surfaces compute EEs in AdS geometries with a thin brane, the remaining step is to take the BCFT limit.

## 5.2 Quantum Extremal Surfaces for BCFTs

As in section 4.1, we can now try to interpret the BCFT limit of the three-dimensional geodesic as a QES in the "intermediate" 2d braneworld picture in the large-$T$ limit. This section mirrors 4.1 very closely, so we will be brief. We will also take

$$c_{\mathrm{II}} = c \,, \qquad \sigma_{\mathrm{II}} = \sigma \,, \qquad \epsilon_{\mathrm{II}} = \varepsilon \,, \tag{5.8}$$

for notational convenience. As before, we view the effective action on the brane as resulting from integrating out bulk degrees of freedom up to the brane [74] (similar ideas have been discussed in [14, 15, 48, 75, 76]), and the generalized entropy is the entropy of this weakly gravitating two-dimensional conformal theory, where we take the UV cutoff along the brane insertion to be $y$ dependent $\varepsilon_{\mathrm{brane}}(y) = y \cos \psi_{\mathrm{II}}$ given the angle the brane makes with the boundary at $z = 0$. We find

$$S_{\mathrm{gen}}(y) = \frac{c}{6} \log \left[ \frac{(\sigma + y)^2}{y \varepsilon} \frac{1}{\cos(\psi_{\mathrm{II}})} \right] \,. \tag{5.9}$$

The effect of gravity on the brane is accounted for by requiring that the quantum extremal surface be an extremum over all possible island locations $y$ in (5.9). Thus

$$\partial_y S_{\mathrm{gen}} = 0 \qquad \Rightarrow \qquad y^* = \sigma \,. \tag{5.10}$$

Notice that the position of the QES matches exactly the point where Takayanagi's geodesic meets the EOW brane [17, 47]. The entropy of the island saddle in the intermediate picture then reads

$$S_{\mathrm{island}} = \frac{c}{6} \log \left[ \frac{2\sigma}{\varepsilon} \frac{2}{\cos(\psi_{\mathrm{II}})} \right] \,. \tag{5.11}$$

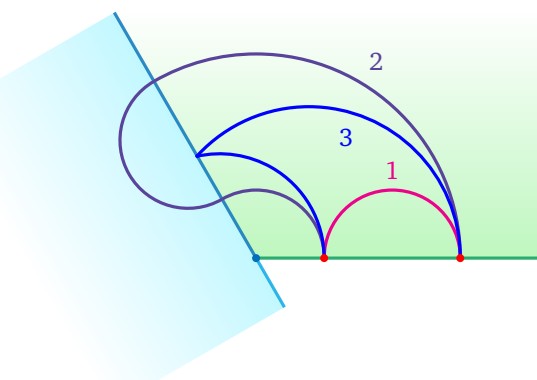

Figure 14: The various geodesic saddles in BCFT, recovered from the $\nu \to 0$ limit of ICFT.

We can compare this result with the one obtained using Takayanagi's prescription in three dimensions,

$$S_{\text{RT}}^{\text{3D}} = \frac{c}{6} \log \left[ \frac{2\sigma}{\varepsilon} \tan \left( \frac{\psi_{\text{II}}}{2} + \frac{\pi}{4} \right) \right]. \tag{5.12}$$

To see that these match in the large-tension limit, recall that $T_{\text{max}}$ is reached by taking $\eta \to 1$ in (5.2)[14]. Parametrizing $\eta = 1 - \delta^2/2$, this translates to $\psi_{\text{II}} = \frac{\pi}{2} - \delta + \mathcal{O}(\delta^3)$ as $\delta \to 0$. Plugging this into the above formulas, we find

$$S_{\text{island}} = -\frac{c}{6} \log(\delta) + \frac{c}{6} \log \left[ \frac{4\sigma}{\varepsilon} \right] + \mathcal{O}(\delta^2), \tag{5.13}$$

$$S_{\text{RT}}^{\text{3D}} = -\frac{c}{6} \log(\delta) + \frac{c}{6} \log \left[ \frac{4\sigma}{\varepsilon} \right] + \mathcal{O}(\delta^2), \tag{5.14}$$

where we find a mismatch between these formulas only at $\mathcal{O}(\delta^2)$. The match between the QES formula and the RT calculation is somewhat cleaner in the BCFT limit as compared to ICFT, since the location of the island agrees for any value of the tension, although the entropies still only match in the limit of $T \to T_{\text{max}}$.

It is curious to remark that a different choice of position dependent cutoff, namely

$$\varepsilon_{\text{brane}}(y) = \frac{2y}{\tan \left( \frac{\psi_{\text{II}}}{2} + \frac{\pi}{4} \right)}, \tag{5.15}$$

would have resulted in an *exact* match between the island prescription and the three-dimensional RT formula, valid for all values of the tension $T$. While this would be appealing, we have no geometric interpretation of such a formula and moreover do not expect the theory on the brane to be locally conformal away from the large-$T$ limit. We thus cannot justify using the CFT entropy formula when computing the generalized entropy for arbitrary $T$.

### 5.3 BCFT two-point functions

Let us now analyze the BCFT limit of geodesics connecting two points on the same CFT (figure 6), where we take the insertions on the side whose central charge we are keeping finite. We readily see that our limit gives the structure of geodesics in AdS/BCFT studied in [58]. We label the possible RT geodesics by (1), (2) and (3) in figure 14, where (1) is the standard

---

[14]The numerical bootstrap might be able to put bounds on $\eta$: see [77] for a proof of concept.

"connected" RT surface whereas (2) and (3) are "disconnected" surfaces that can end on the EOW brane. Based on our ICFT setup, we expect (2) and (3) to be related to the two solution branches labeled $s_1^*$ and $s_2^*$ in section 3.4.3.

What happens as $v \to 0$ in the ICFT setup is that surfaces of type (3) probe the region behind the brane less as $v$ is decreased, and ultimately reflect off the EOW brane. This is precisely the structure described in [58]. Let us return to (3.67) in the BCFT limit. Now we must solve:

$$\Theta = \frac{\sigma_{\mathrm{II}}^P}{\sigma_{\mathrm{II}}^Q} = \frac{s(s+2) + 2\sin(\psi_{\mathrm{II}})\left(\sin(\psi_{\mathrm{II}}) + \sqrt{s(s+2) + \sin^2(\psi_{\mathrm{II}})}\right)}{s^2} . \qquad (5.16)$$

The solutions that describe geodesics of type (2) are given by $s_2^* \to \infty$. Indeed in the $v \to 0$ limit, the red curve of figure 7 is pushed upwards towards infinity. The solutions that describe curves of type (3) on the other hand are given by

$$s_1^* = \frac{2}{\Theta - 1}\left(1 + \sin(\psi_{\mathrm{II}})\sqrt{\Theta}\right) , \qquad (5.17)$$

however this only satisfies (5.16) for

$$\sin(\psi_{\mathrm{II}}) > -\frac{2\sqrt{\Theta}}{1 + \Theta} . \qquad (5.18)$$

We also remind the reader that

$$s > \begin{cases} 0 , & \psi_{\mathrm{II}} \in [0, \pi/2], \\ -1 + \cos(\psi_{\mathrm{II}}) , & \psi_{\mathrm{II}} \in [-\pi/2, 0]. \end{cases} \qquad (5.19)$$

We see that, for negative tension, *i.e.* $\psi_{\mathrm{II}} < 0$, when $\Theta$ is greater than some critical value set by $\psi_{\mathrm{II}}$, reflecting geodesics of type (3) cease to exist. In this case geodesics of type (1) also do not exist and we are left with the "disconnected" geodesics of type (2). Thus, in a rather intuitive way, starting from ICFT, we have recovered some of the results in [58].

## 6 Conclusions

In this paper we have established three interrelated goals. We established in detail how double holographic setups, and in particular ones of interest in black hole evaporation scenarios, can be constructed starting from holographic interface conformal field theories. For this construction one needs to adopt several – ultimately equivalent – points of view. A major advantage of our approach is that it operates squarely within the confines of the standard rules of $\mathrm{AdS}_{d+1}/\mathrm{CFT}_d$, making computations very transparent. From this perspective our holographic boundary system consists of two (potentially different) CFTs, which are coupled via a shared interface along a common hypersurface of one lower spatial dimension than each of the two constituent $\mathrm{CFT}_d$. However, as we explained, the conformal interface itself hosts degrees of freedom, which interact with both of the adjacent $\mathrm{CFT}_d$. The combined system, by assumption, has a holographic bulk dual of dimension $d + 1$, which encodes the kinematics and dynamics of the $\mathrm{ICFT}_d$ on the boundary. In this paper we used a thin-brane approximation in order to extend the interface into the holographic bulk, and it would clearly be interesting to understand the consequences of relaxing this assumption in future work—see *e.g.* [78] for some steps in this direction. Indeed, explicit top down models of interface CFTs often underline the necessity of relaxing the thin-brane setup [79–83], while models closer to the setup considered

here might be possible [84,85], but fine-tuned [86]. The point of view most directly related to black hole evaporation is the double holographic one, where we actually think of the interface degrees of freedom as its own boundary theory, $I_{d-1}$, with a $d-$dimensional gravitational bulk confined to the world-volume of the brane separating the holographic duals of both boundary $CFT_d$. The interactions between the interface and the $CFT_d$ are now reinterpreted as coupling the interface to a bath (see figure 2 as a reminder), the conceptual key to enabling the evaporation of black holes that may exist in the the gravitational dual of the interface degrees of freedom. In other words, we have a holographic dual of an open quantum system, which may host evaporating black hole configurations. While our setup in principle can be realized for general $d$, for the most part our results are in $d = 2$, where further simplifications arise for example in computing entanglement entropies via twist operators. It would clearly be of interest to generalize these results to higher dimensions. However, it is important to remark that our results indicate that qualitative features will likely remain the same, as it was found, for example, in [11]. In higher dimensions, tracing out the interface and part of the baths will demand a holographic computation of the area of an entangling surface that crosses the brane. What remains to show is that there exists an "intermediate" weakly gravitating description localized at the brane whose generalized entropy matches that of the brane-crossing RT surface, and that the location of the island matches the crossing region of this RT surface.

Exploiting this straightforward realization of holographic system + bath setups, we computed the entanglement entropy of the Hawking radiation in terms of a conventional description. This means that we rely only on the $d + 1$ dimensional holographic dual, and employ a standard (H)RT computation of the entanglement entropy, [65, 66, 87] between the interface degrees of freedom and (a part of) the bath. The structure of this computation parallels the semiclassical 'Hawking' and 'replica' saddle pattern [6, 9, 69], identifying the transition between the two as a more familiar phase transition between different minimizers of the RT surface in the $AdS_{d+1}$ setup. This realization, however, allows us to go further, and in particular we proved that this same phase transition, as seen from the double-holographic perspective is precisely the phase transition seen in the generalized entropy formula applied to the two-dimensional black hole on the bulk brane(-world). Given that the entire argument follows from standard (H)RT rules, one may regard this as a proof of the island formula, at least in double holography. This further allowed us to take the BCFT limit (either $c_I/c_{II} \to 0$ or $c_{II}/c_I \to 0$ as explained in section 5) of our results and to give a derivation of the holographic BCFT prescription to calculate entanglement entropy. Put together with our arguments in the original ICFT setup, this implies that the recent work on islands from the BCFT perspective follows from a proven tool in holography [65].

However, interesting points on consistency of islands in theories with long-range gravity have been raised in [13, 88]. In particular, it has been argued that in general dimensions $d$ the gravitons on the brane are massive. The case explicitly worked out in this paper ($d = 2$) does not have gravitons on the brane, but the point still holds: the coupling of the theory on the brane with the baths implies that the stress–energy tensor on the brane is not conserved, which breaks gauge invariance. This can also be seen from a purely boundary perspective: for conformal boundary conditions the boundary spectrum of a CFT does not have a spin-2 primary with dimension $d - 1$. For $d = 2$ the story is slightly different (see [35] for instance), but the outcome is similar: the boundary theory cannot have a well-defined stress tensor. In this respect the present work does not bring any advances to the understanding of these issues. A related extension of our setup might instead be possible and interesting: while we cannot couple the system to a bath and maintain a conserved stress tensor, it is at least possible to devise couplings where the interface can store energy. These non-conformal interfaces allow for long-lived bound states possibly dual to a more realistic evaporating black hole.

We conclude this paper by giving an outlook as well as mentioning some open ends we

would like to come back to in the future. Firstly, the geodesic approximation for correlation functions in the holographic bulk of ICFTs we determined here should have interesting detailed implications for the (boundary) OPE of its holographic dual. It would be interesting to carry out this expansion in future work, and to potentially use it to constrain the properties that distinguish holographic ICFTs from their generic relatives. Recently, [86] derived constraints for the applicability of a thin-brane description in holographic ICFT, along the lines of the bulk-point approach initiated in [89, 90], and it would be interesting to see what can be learned from comparing to the correlation functions we have derived here.

In this paper we have for the most part investigated the case where the boundary ICFT is prepared in the thermofield double state. As we argued, this means that the 2D bulk braneworld blackhole is an eternal (two-sided) geometry (see figure 11). It would be very interesting to also consider the case of one-sided black holes which form from collapse (see e.g. [91–94]). It is tempting to speculate that such scenarios can be engineered by considering the solutions of [44] away from equilibrium.

## Acknowledgements

We would like to thank Costas Bachas, Ivano Basile, Lorenzo Bianchi, Nikolay Bobev, Shira Chapman, Ben Craps, Stefano De Angelis, Oleksandr Gamayun, Raghu Mahajan, Vassilis Papadopoulos, Joao Penedones, Edgar Shaghoulian and Manus Visser. This work has been supported in part by the Fonds National Suisse de la Recherche Scientifique (Schweizerischer Nationalfonds zur Förderung der wissenschaftlichen Forschung) through Project Grants 200020_182513, the NCCR 51NF40-141869 The Mathematics of Physics (SwissMAP), and by the DFG Collaborative Research Center (CRC) 183 Project No. 277101999 - project A03. T.A. is supported by the Delta ITP consortium, a program of the Netherlands Organisation for Scientific Research (NWO) that is funded by the Dutch Ministry of Education, Culture and Science (OCW). M.M is supported by the Swiss National Science Foundation through the Ambizione grant number 193472.

## A   Length of the geodesic with points on the same side

In this section we compute the length of the geodesic in figure 6, using the notation of section 3.4.3. Here we assume that values of the parameters allow for the existence of a geodesic. The geodesic is divided into three arcs, so the length can be expressed as:

$$d(\sigma_{\mathrm{II}}^Q, \sigma_{\mathrm{II}}^P) = d(Q,V) + d(V,S) + d(S,P) \,. \tag{A.1}$$

We will compute all the three pieces independently. Taking inspiration from (3.48), it's not hard to conclude that

$$d(Q,V) = L_{\mathrm{II}} \log\left[\frac{2\,\overline{CQ}}{\varepsilon_Q} \tan\left(\frac{\alpha}{2}\right)\right] \,, \tag{A.2}$$

and

$$d(S,P) = L_{\mathrm{II}} \log\left[\frac{2\,\overline{AP}}{\varepsilon_P} \tan\left(\frac{\gamma}{2}\right)\right] \,, \tag{A.3}$$

where $\varepsilon_Q$, $\varepsilon_P$ are (possibly different) cutoffs at $Q$ and $P$. On the other hand $d(V,S)$ is cutoff independent, since we can express it as the difference of two circular arcs that start at the

same point on the (continuation behind the interface of the) conformal boundary I, thus:

$$d(V,S) = L_{\text{I}} \log \left[ \frac{\tan\left(\frac{\widehat{OBS}}{2}\right)}{\tan\left(\frac{\widehat{OBC}}{2}\right)} \right] . \tag{A.4}$$

All we need are the various angles subtended by the arcs. Looking at triangle $\triangle OBC$ and recalling $\widehat{AOB} = \psi_{\text{I}} + \psi_{\text{II}}$, we find

$$\widehat{OBC} = \psi_{\text{I}} + \psi_{\text{II}} - \alpha . \tag{A.5}$$

On the other hand, looking at triangle $\triangle SBV$ (and using (3.60)) we obtain

$$\widehat{SBV} = 2(\alpha - \psi_{\text{II}}) , \tag{A.6}$$

meaning

$$\widehat{OBS} = \widehat{SBV} + \widehat{OBC} = \alpha - \psi_{\text{II}} + \psi_{\text{I}} , \tag{A.7}$$

thus

$$d(V,S) = L_{\text{I}} \log \left[ \frac{\tan\left(\frac{\alpha - \psi_{\text{II}} + \psi_{\text{I}}}{2}\right)}{\tan\left(\frac{\psi_{\text{I}} + \psi_{\text{II}} - \alpha}{2}\right)} \right] . \tag{A.8}$$

Now considering equations (3.64, 3.63) we obtain

$$\overline{CQ} = \sigma_{\text{II}}^{Q} + \lambda = \overline{CV} = \lambda \frac{\cos(\psi_{\text{II}})}{\cos(\alpha - \psi_{\text{II}})} , \tag{A.9}$$

$$\overline{AP} = \sigma_{\text{II}}^{P} - \overline{AO} = \overline{AS} = \lambda \frac{\cos(\psi_{\text{II}}) \sin(\alpha) \sin(\alpha + \psi_{\text{I}} - \psi_{\text{II}})}{\sin(\alpha - 2\psi_{\text{II}}) \sin(\psi_{\text{I}} + \psi_{\text{II}} - \alpha) \cos(\alpha - \psi_{\text{II}})} . \tag{A.10}$$

Then, using the relation (3.61) between $\gamma$ and $\alpha$ and putting everything together, we obtain

$$d(\sigma_{\text{II}}^{Q}, \sigma_{\text{II}}^{P}) = L_{\text{II}} \log \left[ \frac{2\overline{CQ}}{\varepsilon_{Q}} \tan\left(\frac{\alpha}{2}\right) \right] + L_{\text{II}} \log \left[ \frac{2\overline{AP}}{\varepsilon_{P}} \cot\left(\frac{\alpha - 2\psi_{\text{II}}}{2}\right) \right]$$
$$+ L_{\text{I}} \log \left[ \tan\left(\frac{\alpha - \psi_{\text{II}} + \psi_{\text{I}}}{2}\right) \cot\left(\frac{\psi_{\text{I}} + \psi_{\text{II}} - \alpha}{2}\right) \right] . \tag{A.11}$$

What remains is to invert (3.64) to obtain $\lambda(\sigma_{\text{II}}^{Q}, \sigma_{\text{II}}^{P})$ and $\alpha(\sigma_{\text{II}}^{Q}, \sigma_{\text{II}}^{P})$ using the strategy outlined in section 3.4.3.

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
