# Peer review of "Sailing past the End of the World and discovering the Island"

_SciPost Physics, doi:SciPost Phys. 13, 075 (2022)_

## Round 2 · Referee Report · Anonymous (Referee 1) · 2022-5-31

Strengths

1 - Gives intuitive arguments for prescriptions to calculate holographic BCFT and island contributions to entanglement
2 - Arguments rely on simple geometry
3 - Well written and pedagogical
4 - Clear figures

Weaknesses

1 - Somewhat minimal variation on existing work in double holographic methods for entanglement entropies
2 - Restricted to 2D holographic CFTs and thin brane approximation

Report

In this work, the authors construct holographic duals to two-dimensional “interface CFTs,” (ICFTs) which describe the coupling of two CFTs with large and potentially different central charges at an interface or defect. The gravitational picture is that of two patches of AdS_3 with potentially different radii (identified with the two different Brown-Henneaux central charges) which meet along a brane described by standard Israel-Lanczos matching conditions. These conditions lead to a transparent interface which allows for non-zero correlations between operators in the separate CFTs. The authors then proceed to describe a procedure for obtaining two-point functions of heavy operators and the entanglement entropy of subregions of the ICFTs, both of which are described by geodesics in the bulk. The necessary criteria they derive for such geodesics is that they must be continuous and normal to the brane at the bulk AdS interface. By then considering certain limits and interpretations of this setup, the authors explain the appearance of islands in the computation of the generalized entropy in AdS_2 and the resulting Page curve for evaporating black holes as well as the procedure for obtaining the entanglement entropy in a holographic BCFT. The most important point of the work is that the criteria for the geodesics with a thin brane interface follows from smoothness of the bulk and the well-established HRT procedure for entanglement entropy.

I find the paper to be well written and pedagogical. Furthermore, the use of AdS_3 allows the geodesics to be obtained from simple geometry, making the results clear and unobjectionable.

The obvious drawback, which the authors emphasize, is that AdS_3 is a special case of holography, where, in particular, some objections due to massive gravitons and braneworlds cannot be addressed. Nevertheless, because the results of the paper rely mostly on the HRT procedure, for which a proof exists in the literature, it is likely some version of their construction continues to hold in higher dimensions. This is important evidence for the applicability of the so-called island formula in higher dimensions. Another drawback, as the authors similarly point out, is that the coincidence of the RT procedure and the interface boundary conditions do not allow brane-localized degrees of freedom, though from the CFT point of view, such degrees of freedom should exist generically. Nevertheless, to address this in detail is likely beyond the scope of this work, though if the authors had any speculative ideas in this direction, I would be curious to hear them.

Requested changes

The paper needs some editing for typos and repeated words, for instance:

1 - on p.34, "in the large tension limit" is repeated 2- p.40 "at last" to "at least"

  • validity: high
  • significance: good
  • originality: good
  • clarity: high
  • formatting: excellent
  • grammar: excellent

Author:  Pietro Pelliconi  on 2022-06-28  [id 2612]

(in reply to Report 1 on 2022-05-31)
Category:
answer to question

We thank the referee for their valuable comments and questions, and for pointing out typos. The question about the addition of localized degrees of freedom on the brane is interesting. One option is to add JT gravity on the brane, but this choice is non-unique and, more generally, one could couple gravitationally an interacting 2d quantum field theory. The new degrees of freedom would correspond to local (single and multi-trace) interface operators in the dual ICFT. However, it is worth pointing out that the model analyzed in the paper already contains a non-trivial spectrum of interface operators, whose quantum numbers depend on the brane tension. In this sense, we do not expect the physics to change qualitatively. Certainly, it would be interesting to perform computations in explicit models and explore how the Page curve depends on the additional parameters.

---

## Round 2 · Referee Report · Jorrit Kruthoff (Referee 2) · 2022-6-6

Report

This manuscript considers the calculation of entanglement entropies in two conformal field theories joint through an interface. The interface is permeable and allows for information to be exchanged between the two sides. The authors consider the CFTs to be holographic, which results in a bulk theory that consists of two AdS spaces with different curvature radii separated by a thin brane. One can also consider integrating out the bulk AdS degrees of freedom so that one gets an effective gravitational theory on the brane. Using these three different perspectives, the authors are able to derive the island formula in the perspective with the gravitating brane just using standard rules for AdS/CFT.

The manuscript is clear and contains many explicit calculation. I think it is valuable to see how the island rule emerges from standard rules of AdS/CFT.

I think the manuscript should be published, but I had some questions purely for my own interest and it would be great if the authors could give some comments.

1) In your setup, can I think of the Page transition where the Hawking and replica saddle change dominance as the same as the transition for a two-interval entanglement entropy? If so, I find it amusing to see this so explicitly in your setup.

2) I was a bit confused to what extend you now have a setup that can go beyond the usual island rule because you can just use the standard AdS/CFT rules. Is it that you can consider more general states? It would be great if the authors could comment on this.

3) I know at least a subset of the authors is a de Sitter afficionado, so I was wondering whether this setup can also be used to study islands in cosmology. I guess there the derivation of the RT prescription is already conjectural, but nevertheless it would be interesting to study. Is the bulk now separated by a timelike brane in this case?

  • validity: high
  • significance: high
  • originality: high
  • clarity: top
  • formatting: excellent
  • grammar: excellent

Author:  Pietro Pelliconi  on 2022-06-28  [id 2613]

(in reply to Report 2 by Jorrit Kruthoff on 2022-06-06)

We thank the referee for their valuable comments and questions. We address them in the following, numbering our answers in correspondence with the referee’s questions.

1) This is correct, and indeed it is an important feature of the double holographic realizations of Page curves. In our work, unlike in previous work on the subject, the RT surface after the transition is not fixed by conformal symmetry, or equivalently, by the identity in the OPE. Instead, it contains dynamical information, which is encoded in a non trivial function of a cross ratio.

2) Our main interest in the present paper is to provide a derivation of the island rule in a setup where the symmetries do not fix the result. Therefore, the match between the 2d and the 3d computations lends strong support to the rule as it is. Nevertheless, the referee is correct that the model might allow to inquire about possible corrections to the island rule. Indeed, the brane tension parametrizes the strength of the gravitational coupling on the brane, while the 3d theory is semi-classical for all values of the tension. Trying to reproduce the 3d computation in the 2d perspective, when tuning the tension away from the critical value, could teach us more about entanglement in gravity.

3) To our knowledge, this has not been explored much in the literature, but it might be possible. In particular, an interesting feature of the brane-world models is that the geometry on the brane can be (global) de Sitter, for appropriate (beyond critical) values of the tension and Lorentzian bulks. An alternative avenue might be to try to embed the centaur geometry of Anninos and Hofman in the ICFT framework, where we have in mind (only) the static patch of de Sitter living on the brane. Since this would require a drastic change to the setup discussed in our paper, we leave this possibility to future work.

---

## Round 3 · Referee Report · Anonymous (Referee 3) · 2022-7-8

Strengths

  1. Results are new and interesting.
  2. Clearly written and coherently presented.

Weaknesses

I believe that the authors could have a better understanding of existing literature.

Report

This paper is interesting, it considers holographic interface conformal field theory (ICFT) and its reduction to holography boundary conformal field theory (BCFT). The model is described by braneworld and it is doubly holographic. The authors exploited the standard three equivalent descriptions of the braneworld holography to compute the correlators in the boundary description and look for its interpretation in the intermediate description regarding black holes. The calculation is done in the bulk description by geodesics approximation. There are two new perspectives from this paper. This first is that the geodesics computing the single sided ICFT correlator can go across the brane and come back and the second is that the island rule in the BCFT case can be derived from the ICFT case. This first perspective is interesting but in this paper the authors exclusively considered the AdS3 case where the geodesics are very simple and easy to cut and glue. It would be interesting to see whether the brane crossing and coming back geodesics exist in higher dimensional holographic ICFTs. The second perspective simply exploits the fact that the brane is gravitating and when we minimize the area of the bulk Ryu-Takayanagi surface we should also minimize over the possible end points of the bulk Ryu-Takayanagi surface on the brane. I recommend SciPost to accept this paper for publication.

Requested changes

I believe that the authors could have a better understanding of existing literature. Regarding this point, here are a few suggestions:

  1. On top of Page 3: For BCFT models that can perform studies of black hole coupled to bath add https://inspirehep.net/literature/1751747, https://inspirehep.net/literature/1766462 and https://inspirehep.net/literature/1799453 to [9-12] and organize them chronologically. (The three equivalent descriptions are firstly emphasized in https://inspirehep.net/literature/1799453.)

  2. Add https://inspirehep.net/literature/1766462 and https://inspirehep.net/literature/1799453 to [9,11] in “ Recent progress in understanding black-hole evaporation came from computing the same observables in either of the three descriptions”.

  3. The relationship between BCFT entanglement entropy and island rule in braneworld holography is firstly refined in https://inspirehep.net/literature/1804322 and https://inspirehep.net/literature/1835408 (see equ (2.1)-(2.2)).

---

## Editorial Decision

published